# A general strategy for heterogenizing olefin polymerization catalysts and the synthesis of polyolefins and composites

Chen Zou [1], Guifu Si[1] & Changle Chen [1 ✉]

The heterogenization of homogeneous metal complexes on solid supports presents an efficient strategy for bridging homogeneous catalysts with industrially-preferred heterogeneous catalysts; however, a series of drawbacks restrict their implementation in olefin polymerization, particularly for copolymerization with polar comonomers. In this contribution, we report an ionic anchoring strategy that is highly versatile, generally applicable to different systems, and enables strong catalyst-support interactions while tolerating various polar functional groups. In addition to greatly enhanced polymerization properties, the supported catalysts achieved higher comonomer incorporation than their unsupported counterparts. This strategy enabled efficient polymerization at high temperatures at large scale and great control over product morphology, and the facile synthesis of polyolefin composites. More importantly, the dispersion of different fillers in the polyolefin matrix produced great material properties even at low composite loadings. It is expected that this strategy will find applications in different catalytic systems and the synthesis of advanced engineering materials.

[1] CAS Key Laboratory of Soft Matter Chemistry, Department of Polymer Science and Engineering, University of Science and Technology of China, Hefei 230026, China. ✉email: changle@ustc.edu.cn

The annual production of synthetic plastics exceeds 380 million tonnes, with polyolefins representing half of this amount[1,2]. It has been more than fifty years since Ziegler and Natta won the Nobel Prize for their contributions to olefin polymerization catalysts. Over the past few decades, the design and development of high-performance catalysts have received tremendous attention from both academia and industry[3–5]. The introduction of polar-functional groups into the otherwise nonpolar backbone of polyolefins could further broaden their applications in many areas[6]. Transition-metal-catalyzed copolymerization of olefins with polar comonomers holds great potential for accessing polar functionalized polyolefins[7–9], and has been recognized as one of the last holy grails in this field[10]; however, this method has not yet been applied in the industry, despite decades of research. Industrially successful early transition-metal catalysts cannot perform such tasks due to the poisoning effects of polar groups towards metal centers. Within this context, late-transition-metal-based olefin polymerization catalysts have received continuous interest because of their low oxophilicity and correspondingly high tolerance towards polar groups[11–18].

The polyolefin industry mainly utilizes heterogeneous systems due to their ability to control product morphologies, which enables continuous polymerization processes and prevents reactor fouling[19–21]. In contrast, homogeneous systems possess the advantages of defined molecular structures and rational modification, which make them useful for mechanistic studies. The surface organometallic community originated in the 1970s to understand the exceptional activity of Ziegler-Natta compositions relative to organometallics known at the time[22]. The heterogenization of homogeneous metal complexes on solid supports through surface organometallic(/coordination) chemistry (SOMC)[23,24] has been extensively studied for organic transformations and has led to the discovery of many superior catalytic systems[25,26]. This concept is fascinating for polyolefin research because it provides "drop-in" catalyst solutions for existing technologies in addition to combining the advantages of both types of catalysts. Many heterogenized systems based on early transition-metal catalysts have already been successfully commercialized[27,28].

There are three major routes for the heterogenization of olefin polymerization catalysts on solid supports: (Fig. 1, Route a) the introduction of a precatalyst to a cocatalyst-pretreated solid support is the most popular strategy; (Route b) the introduction of a cocatalyst to a precatalyst-pretreated solid support is usually avoided due to potential side reactions between the precatalyst and solid support; (Route c) activation of the precatalyst with a cocatalyst prior to impregnation of the solid support is not suitable for systems with sensitive active species. These commonly used routes suffer from a series of drawbacks, including catalyst leaching due to weak interactions between the active metal species and solid support. More importantly, these drawbacks will be dramatically magnified in the presence of polar comonomers. For example, a large amount of expensive and sensitive aluminum cocatalyst is usually required, which poses the additional issue of side reactions with polar comonomers. The interactions of the active site with the support play important roles in determining the properties of the catalysts[29]. However, these interactions may be significantly weakened or altered by Lewis basic polar-functional groups. The covalent tethering of a precatalyst to a solid support can address these issues[30], but it is of little practical value because it suffers from complicated syntheses, as well as difficult in catalyst characterization.

Despite the existence of hundreds of potent homogeneous catalysts in the literature after decades of research, very few late-transition-metal-based heterogeneous systems have been investigated for ethylene polymerization[31–35]. Among them, even fewer studies investigated the copolymerization of ethylene with polar

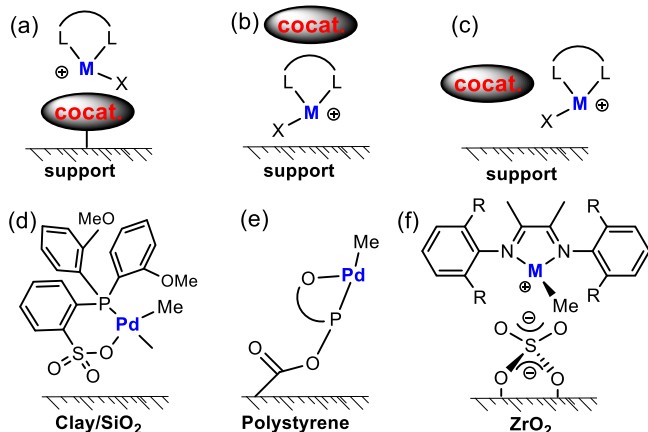

**Fig. 1 Commonly used heterogenization strategies and examples of known heterogeneous late-transition metal catalysts. a–c** Commonly used heterogenization strategies. **d–f** Examples of known heterogeneous late-transition metal catalysts for the copolymerization of ethylene with polar comonomers.

comonomers. In 2014, Mecking and coworkers reported the synthesis of clay or silica-supported phosphine-sulfonate palladium systems (Fig. 1d) and demonstrated the first example of ethylene-acrylate insertion copolymerization using supported solid catalysts. This system showed moderate activity ($142 \, \text{kg mol}^{-1} \text{h}^{-1}$), moderate comonomer incorporation (up to 4.3%), and low copolymer molecular weight ($M_n = 4700 \, \text{g/mol}$)[36]. The covalent binding of this type of palladium catalyst to functionalized polystyrene significantly decreased the activity and comonomer incorporation, likely due to limited diffusion of the monomer and polymer product in the polystyrene support (Fig. 1e). Conley and coworkers supported an α-diimine nickel complex on sulfated zirconium oxide (Fig. 1f). This catalyst mediated ethylene copolymerization with methyl 10-undecenoate with a low activity (ca. $4 \, \text{kg mol}^{-1} \text{h}^{-1}$), leading to the formation of a copolymer with a moderate molecular weight ($M_n = 23600 \, \text{g/mol}$; Đ = 4.15) and low comonomer incorporation (0.4%)[37]. A similar palladium system copolymerized ethylene with methyl acrylate at a low activity ($14.9 \, \text{kg mol}^{-1} \text{h}^{-1}$), affording a copolymer with a moderate molecular weight ($M_n = 16000 \, \text{g/mol}$; Đ = 4.15) and low incorporation (0.46%)[38]. Cai and coworkers prepared a heterogeneous system through the reaction of an anilinonaphthoquinone nickel complex with methylaluminoxane (100 equiv.) modified silica (MMAO/SiO$_2$), which could copolymerize ethylene with 5-hexene-1-yl acetate and allyl acetate[39,40].

In this contribution, we design an ionic anchoring strategy (IAS) for the heterogenization of various transition metal catalysts. This strategy is highly versatile, generally applicable to different systems, and most importantly, enables strong catalyst-support interactions that can tolerate polar-functional groups. Furthermore, polyolefins and composites with great material properties can be easily accessed using this strategy.

## Results and discussion

**Synthesis and characterization of the nickel and titanium complexes**. The hydroxyl moiety can be introduced to various ligands through simple reactions starting with commercially available regents (Fig. 2). The two hydroxyl groups of commercially available 2-$t$-butylhydroquinone were protected with tetrahydro-2H-pyran. The subsequent lithiation and reaction with (2',6'-dimethoxybiphenyl-2-yl)phenylphosphine chloride, followed by deprotection, formed the hydroxyl-functionalized phosphino-phenol ligand. The reaction of this ligand with the

**Fig. 2 Synthesis of hydroxyl-functionalized ligands and the corresponding metal complexes. a** Phosphinophenolate nickel. **b** Perfluorophenyl-substituted phosphinophenolate nickel. **c** Phenoxy-imine titanium.

precursor (pyridine)$_2$NiMe$_2$ generated **Ni-OH**. Subsequent reaction with NaH afforded the desired nickel complex (**Ni-ONa**) in near-quantitative yield. Inspired by Mecking and coworkers' recent works on perfluorophenyl-substituted phosphinophenolate nickel catalyst[11], the corresponding OH and ONa tagged nickel catalysts (**Ni-F-OH** and **Ni-F-ONa**) were also prepared. $^1$H NMR characterization showed the coexistence of Ni-*Me* and Ar-*OH* signals in both **Ni-OH** and **Ni-F-OH**, and the integration of Ni-*Me* signal matched very well with the rest of the ligand signals (Supplementary Fig. 1). Furthermore, the addition of *p*-cresol to nickel complex without the OH group (**Ni**) indicated no reaction at all (Supplementary Fig. 2). The hydroxyl-functionalized salicylaldimine ligand was prepared from the condensation of commercially available salicylaldehyde with 4-amino-phenol. The subsequent reaction of this ligand with 2 equiv. of NaH and 0.5 equiv. of TiCl$_4$ formed the desired titanium complex (**Ti-ONa**) in high yield. For these metal complexes, their analogs without the ONa tag (**Ni** and **Ti**) were prepared for comparison. Notably, the synthesis of these three metal complexes used hydroxyl-bearing commercially available compounds. Simple reactions such as condensation or lithiation were employed and afforded the desired products in high yields. Furthermore, the presence of a hydroxyl or ONa group did not interfere with the metal complexation reaction. Most importantly, sodium could be easily substituted with other metals (Li, K, Ca, etc.) and it is possible to install the hydroxyl group at different positions on the ligand framework.

**Synthesis of the SiO$_2$ supported catalysts**. The supported catalysts were prepared by stirring the complex solution with the solid support. The maximum adsorption capacity of **Ni-ONa** on SiO$_2$ was determined to be 77 μmol cat./1 g SiO$_2$, which is approximately 1.9 times higher than that of **Ni-OH** and 6 times higher than that of **Ni** (Supplementary Fig. 3a). Similarly, the maximum adsorption capacity of **Ti-ONa** on SiO$_2$ is much higher than its corresponding counterparts (**Ti-OH/Ti**; Supplementary Fig. 3b). This provides supporting evidence for the strong interaction of the ionic tag with the surface of solid support.

**Ethylene polymerization studies by nickel complexes**. For subsequent polymerization and copolymerization studies, industrially preferred hydrocarbon solvents (hexanes or *n*-heptane) were employed for all systems. All nickel complexes were highly active in ethylene polymerization without the addition of cocatalyst or scavenger. **Ni-ONa** demonstrated much higher activity and a higher polyethylene molecular weight than **Ni-OH** or **Ni** (Table 1, entries 1–3). The utilization of a potassium salt (**Ni-OK**) further increased the activity (Table 1, entry 4). Interestingly, the supported catalyst **Ni-SiO$_2$** showed no activity towards ethylene polymerization (Table 1, entry 5). It is possible that the interaction of the nickel center with the silica surface leads to steric congestion or poisoning of the nickel center. In direct contrast, the supported catalyst **Ni-ONa-SiO$_2$** showed significantly enhanced activity and polymer molecular weight (Table 1, entry 6).

**Active site counting**. Active site counting was performed by measuring the polymerization activity with the addition of a sub-equivalent amount of PMe$_3$ to poison the nickel center. The activities were plotted as a function of the added poison, and the linear least-squares best fit line was extrapolated to the zero activity point. The results showed that approximately 95% of the nickel center was active during ethylene polymerization

**Table 1 Ethylene polymerization studies with the Ni catalysts[a].**

| Ent | Cat. | Ni /μmol | P /atm | T /°C | Yield /g[b] | Act. [b] $(10^6)$ | $T_m$[c] /°C | $M_n$[d] $(10^4)$ | $M_w/M_n$[d] |
|---|---|---|---|---|---|---|---|---|---|
| 1 | Ni | 0.5 | 8 | 80 | 0.52 | 6.2 | 135.9 | 19.4 | 2.4 |
| 2 | Ni-OH | 0.5 | 8 | 80 | 0.68 | 8.2 | 136.2 | 29.2 | 2.3 |
| 3 | Ni-ONa | 0.5 | 8 | 80 | 1.01 | 12.1 | 137.3 | 58.4 | 2.3 |
| 4 | Ni-OK | 0.5 | 8 | 80 | 1.35 | 16.2 | 138.0 | 72.4 | 2.9 |
| 5 | Ni-SiO$_2$ | 0.5 | 8 | 80 | trace | | | | |
| 6 | Ni-ONa-SiO$_2$ | 0.1 | 8 | 80 | 0.51 | 30.6 | 138.5 | 94.4 | 2.1 |
| 7 | Ni-ONa-TiO$_2$ | 0.1 | 8 | 80 | 0.63 | 37.8 | 138.1 | 113.4 | 2.2 |
| 8 | Ni-ONa-Al$_2$O$_3$ | 0.1 | 8 | 80 | 0.78 | 46.8 | 139.0 | 143.9 | 2.7 |
| 9 | Ni-ONa-ZnO | 0.1 | 8 | 80 | 0.80 | 48.0 | 138.9 | 155.0 | 2.5 |
| 10 | Ni-ONa-MgO | 0.1 | 8 | 80 | 0.88 | 52.8 | 139.7 | 241.0 | 2.3 |
| 11 | Ni-ONa-MgO | 0.1 | 30 | 80 | 1.61 | 96.6 | 139.2 | 256.9 | 2.5 |
| 12 | Ni-ONa-MgO | 0.1 | 30 | 30 | 0.35 | 21.0 | 139.9 | 359.5 | 2.4 |
| 13 | Ni-F-ONa-MgO | 0.1 | 8 | 80 | 1.03 | 61.8 | 138.5 | 264.0 | 3.0 |
| 14 | Ni-F-ONa-MgO | 0.1 | 30 | 80 | 1.82 | 109.2 | 138.2 | 326.5 | 2.7 |
| 15 | Ni-F-ONa-MgO | 0.1 | 30 | 30 | 0.32 | 19.2 | 138.9 | 449.2 | 2.0 |

[a]Conditions: The same metal loading (20 μmol/g) was used for all solid supports (10 μmol/g loading for Ni-SiO$_2$); 5 mL n-heptane; time = 10 min.
[b]Polymer yield and activity values are average of at least two runs. Activity = $10^6$ g mol$^{-1}$ h$^{-1}$.
[c]Melting temperature determined by DSC.
[d]$M_n$: $10^4$ g mol$^{-1}$, $M_n$ and $M_w/M_n$ determined by GPC in trichlorobenzene at 150 °C.

(Supplementary Fig. 4), making this system one of the few examples with such capabilities for heterogeneous olefin polymerization catalysts[41,42]. The unique properties of this heterogeneous system may be partially attributed to the strong interactions between the ionic tag with the surface of solid supports.

**Influence of different solid supports**. To make this strategy more versatile, different solid supports were employed, which gave different catalytic activities and generated polyethylenes with different molecular weights and melting points (Table 1, entries 6–10). Narrow polydispersities were obtained with all solid supports, indicating that the single-site characteristics of the nickel complex were maintained after immobilization. Specifically, both the activity and the polyethylene molecular weight followed the order **Ni-ONa-MgO > Ni-ONa-ZnO > Ni-ONa-Al$_2$O$_3$ > Ni-ONa-TiO$_2$ > Ni-ONa-SiO$_2$**. Interestingly, this trend corresponds closely with the order of basicity of these solid supports: MgO > ZnO > Al$_2$O$_3$ > TiO$_2$ > SiO$_2$[43,44]. Furthermore, solid-state $^1$H MAS NMR analysis of these heterogeneous catalysts showed that the resonances of the pyridine moiety gradually shifted downfield in the order: **Ni-ONa-SiO$_2$ > Ni-ONa-Al$_2$O$_3$ > Ni-ONa-MgO** (Supplementary Fig. 5). FT-IR analysis showed that the C-O and the C=N stretching peaks shifted slightly after being supported (Supplementary Fig. 6). These results suggest that the interaction between the more basic and correspondingly more electron-donating solid support with the ONa moiety increased the electronic density of the nickel center. As such, the catalytic properties of the nickel center could be easily tuned over a very wide range by using different solid supports. At higher ethylene pressure, **Ni-ONa-MgO** is able to generate polyethylene with molecular weight of up to $3.595 \times 10^6$ g mol$^{-1}$ (Table 1, entries 11 and 12). Under the same condition, **Ni-F-ONa-MgO** is more active than **Ni-ONa-MgO**, with the capability of producing polyethylene with higher molecular weight (Table 1, entries 13–15). With activities above $10^8$ g mol$^{-1}$ h$^{-1}$ and a polymer molecular weight over $4 \times 10^6$ g mol$^{-1}$, these heterogeneous nickel catalysts are among the most potent nickel-based ethylene polymerization catalysts (Supplementary Fig. 7a, b).

**Ethylene copolymerization studies by nickel complexes**. In addition to their superior properties for ethylene polymerization, these nickel catalysts are also highly active in ethylene copolymerization with a series of fundamental polar comonomers (with

polar groups directly attached to the double bond), allyl polar comonomers, and special polar comonomers (with spacers between the polar groups and double bond). During ethylene-methyl acrylate (E-MA) copolymerization, **Ni-ONa-SiO$_2$** showed a similar activity and copolymer molecular weight, along with lower comonomer incorporation than **Ni-OH** and **Ni-ONa** (Table 2, entries 1–3). The utilization of different solid supports can tune the copolymerization properties. Similar to the above-mentioned ethylene homopolymerization scenario, a more basic solid support generally increased the activity and polymer molecular weight (Table 2, entries 3–7). More importantly, the comonomer incorporation ratios also increased in this order. A more basic solid support produced a more electron-rich nickel center, making it less prone to poisoning by the polar comonomers. Higher ethylene pressure can significantly increase activity and copolymer molecular weight at the expense of comonomer incorporation (Table 2, entry 8). Under the same conditions, **Ni-F-ONa-MgO** is more active than **Ni-ONa-MgO**, generating copolymers with higher comonomer incorporation and slightly lower molecular weight (Table 2, entries 9–11). **Ni-ONa-MgO** can also mediate ethylene copolymerization with other types of acrylate comonomers (Table 2, entries 12–15). Comonomers such as trimethoxyvinylsilane (Table 2, entry 16) and allyl comonomers (Table 2, entries 17 and 18) can also be efficiently copolymerized with ethylene, all with similar properties as in E-MA copolymerization. By decreasing the catalyst loading from 5 μmol to 1 μmol and increasing the ethylene pressure to 30 atm, these heterogeneous nickel catalysts demonstrated activities of up to 4.1 $\times 10^6$ g mol$^{-1}$ h$^{-1}$ in the copolymerization of ethylene with tert-butylacrylate (Table 2, entries 19–21). With the utilization of special polar comonomers bearing long spacers between the double bond and polar groups, very high activities (up to 2.32 $\times 10^6$ g mol$^{-1}$ h$^{-1}$) and high copolymer molecular weights (up to $3.07 \times 10^5$ g mol$^{-1}$) can be achieved (Table 2, entries 22–24). Copolymerization activity and polymer molecular weight usually correlate inversely with polar monomer enchainment. Therefore, it is highly challenging to improve one of the three properties (molecular weight, incorporation, and activity) without sacrificing the other two. Interestingly, supporting the nickel catalysts can simultaneously improve all three parameters (**Ni-ONa-MgO** versus **Ni-ONa**; Table 2, entry 7 versus entry 2). These heterogeneous nickel catalysts are among the most potent catalysts in ethylene copolymerization with simple acrylates

**Table 2 Ethylene copolymerization with Ni catalysts[a].**

| Ent. | Cat. | Comon. | [M]/mol L$^{-1}$ | P /atm | Yield /g[b] | Act.[b] (10$^4$) | Incorp.[c] mol% | $T_m$[d] /°C | $M_n$[e] (10$^4$) | $M_w/M_n$[e] |
|---|---|---|---|---|---|---|---|---|---|---|
| 1 | Ni-OH | MA | 0.1 | 8 | 0.09 | 3.6 | 1.1 | 123.9 | 2.9 | 1.9 |
| 2 | Ni-ONa | MA | 0.1 | 8 | 0.12 | 4.8 | 1.5 | 128.6 | 3.6 | 1.8 |
| 3 | Ni-ONa-SiO$_2$ | MA | 0.1 | 8 | 0.16 | 6.4 | 0.1 | 130.4 | 3.0 | 2.3 |
| 4 | Ni-ONa-TiO$_2$ | MA | 0.1 | 8 | 0.14 | 5.6 | 0.7 | 128.4 | 3.8 | 2.1 |
| 5 | Ni-ONa-Al$_2$O$_3$ | MA | 0.1 | 8 | 0.13 | 5.2 | 1.5 | 124.8 | 3.8 | 1.8 |
| 6 | Ni-ONa-ZnO | MA | 0.1 | 8 | 0.28 | 11.2 | 2.0 | 125.0 | 4.7 | 1.9 |
| 7 | Ni-ONa-MgO | MA | 0.1 | 8 | 0.24 | 9.6 | 2.5 | 124.7 | 5.6 | 2.6 |
| 8 | Ni-ONa-MgO | MA | 0.1 | 30 | 0.65 | 26.0 | 0.8 | 128.6 | 10.1 | 2.6 |
| 9 | Ni-F-ONa-MgO | MA | 0.1 | 8 | 0.31 | 12.4 | 2.9 | 123.3 | 4.6 | 2.1 |
| 10 | Ni-F-ONa-MgO | MA | 0.1 | 30 | 0.76 | 30.4 | 1.3 | 126.2 | 9.3 | 1.8 |
| 11 | Ni-F-ONa-MgO | MA | 0.05 | 30 | 1.14 | 45.6 | 0.5 | 130.8 | 14.7 | 2.9 |
| 12 | Ni-ONa-MgO | (allyl butyl ester) | 0.1 | 8 | 0.19 | 7.6 | 1.5 | 126.5 | 6.2 | 2.3 |
| 13 | Ni-ONa-MgO | (allyl ether ester) | 0.1 | 8 | 0.15 | 6.0 | 1.2 | 125.9 | 3.3 | 1.7 |
| 14 | Ni-ONa-MgO | (allyl tert-butyl ester) | 0.1 | 8 | 0.77 | 30.8 | 0.9 | 128.9 | 14.6 | 1.9 |
| 15 | Ni-ONa-MgO | (allyl tert-butyl ester) | 0.2 | 8 | 0.25 | 10.0 | 1.8 | 126.7 | 6.0 | 1.8 |
| 16 | Ni-ONa-MgO | Si(OMe)$_3$ | 0.1 | 8 | 0.26 | 10.4 | 0.1 | 130.3 | 9.7 | 2.3 |
| 17 | Ni-ONa-MgO | Cl | 0.1 | 8 | 0.06 | 2.4 | 0.3 | 128.2 | 3.8 | 1.8 |
| 18 | Ni-ONa-MgO | CN | 0.1 | 8 | 0.10 | 4.0 | 0.7 | 128.4 | 4.8 | 2.1 |
| 19 | Ni-ONa-MgO | (acrylate tert-butyl ester) | 0.1 | 30 | 0.83 | 166.0 | 0.1 | 133.4 | 83.4 | 3.0 |
| 20 | Ni-F-ONa-MgO | (acrylate tert-butyl ester) | 0.1 | 30 | 0.63 | 126.0 | 1.2 | 128.2 | 11.7 | 2.4 |
| 21 | Ni-F-ONa-MgO | (acrylate tert-butyl ester) | 0.05 | 30 | 2.05 | 410.0 | 0.3 | 132.4 | 34.3 | 4.0 |
| 22 | Ni-ONa-MgO | (long-chain Cl) | 0.5 | 8 | 0.95 | 190.0 | 0.7 | 129.0 | 26.8 | 2.1 |
| 23 | Ni-ONa-MgO | (long-chain OH) | 0.5 | 8 | 1.03 | 206.0 | 0.7 | 129.0 | 29.2 | 2.3 |
| 24 | Ni-ONa-MgO | (long-chain COOMe) | 0.5 | 8 | 1.16 | 232.0 | 0.6 | 130.8 | 30.7 | 2.7 |

[a]Conditions: The same metal loading (20 μmol/g) was used for all solid supports; 5 mL $n$-heptane; time = 30 min; T = 80 °C; Entries 1–18: Ni quantity 5 μmol; Entries 19–24: Ni quantity 1 μmol.
[b]Polymer yield and activity values are average of at least two runs. Activity = $10^4$ g mol$^{-1}$ h$^{-1}$.
[c]Comonomer incorporation ratio was determined by $^1$H NMR in C$_2$D$_2$Cl$_4$ at 120 °C.
[d]Melting temperature determined by DSC.
[e]$M_n$: $10^4$ g mol$^{-1}$, $M_n$ and $M_w/M_n$ determined by GPC in trichlorobenzene at 150 °C.

comonomers as well as special polar comonomers (Supplementary Fig. 7c, d)[45].

**Ethylene polymerization and copolymerization studies by titanium complexes.** Using Et$_2$AlCl as the cocatalyst, the titanium complexes **Ti** and **Ti-ONa** showed high activity during ethylene polymerization, with the ability to generate high-molecular-weight polymer products (Table 3, entries 1 and 2). After being supported on silica, **Ti-SiO$_2$** showed slightly increased activity and polyethylene molecular weight than **Ti** (Table 3, entry 3). In contrast, supporting **Ti-ONa** on silica led to a 4 × increase in activity and a 2 × increase in polyethylene molecular weight (Table 3, entry 4). Similar to the results of the nickel studies, the utilization of a different solid support (Al$_2$O$_3$ and MgO) further increased the

polymerization properties (Table 3, entries 5 and 6). During the copolymerization of ethylene/1-hexene(1-hex), **Ti-SiO$_2$** showed more than an 8-fold decrease (0.3 versus 2.6 mol%) in comonomer incorporation than **Ti** (Table 3, entry 8 versus 7). This may be due to significantly limited comonomer access to the metal center[46]. In contrast, the comonomer incorporation was decreased from 4.0 mol% for **Ti-ONa** to 2.5 mol% for **Ti-ONa-SiO$_2$** (Table 3, entry 10 versus 9). Meanwhile, **Ti-ONa-SiO$_2$** showed a 2 × increase in activity and a 7 × increase in copolymer molecular weight than **Ti-ONa**. Similar trends were observed during the copolymerization of ethylene with 4-methyl-1-pentene (4MP1) (Table 3, entries 11–14).

Based upon the above-mentioned studies on complexes synthesis, heterogenization, catalyst characterization, ethylene

**Table 3 Ethylene polymerization and copolymerization studies with titanium catalysts[a].**

| Ent | Cat. | Comon. | Yield[b]/g | Act.[b] ($10^6$) | Incorp.[c] mol% | $T_m$[d]/°C | $M_n$[e]($10^4$) | $M_w/M_n$[e] |
|---|---|---|---|---|---|---|---|---|
| 1 | Ti | – | 0.26 | 6.2 | – | 136.3 | 51.3 | 2.7 |
| 2 | Ti-ONa | – | 0.35 | 8.4 | – | 136.1 | 138.9 | 2.5 |
| 3 | Ti-SiO$_2$ | – | 0.38 | 9.1 | – | 136.6 | 60.8 | 2.5 |
| 4 | Ti-ONa-SiO$_2$ | – | 1.62 | 38.9 | – | 137.5 | 289.7 | 2.0 |
| 5 | Ti-ONa-Al$_2$O$_3$ | – | 1.71 | 41.0 | – | 137.7 | 325.2 | 1.9 |
| 6 | Ti-ONa-MgO | – | 1.65 | 39.6 | – | 137.2 | 424.2 | 2.7 |
| 7 | Ti | 1-hex | 0.28 | 1.1 | 2.6 | 124.9 | 4.7 | 2.4 |
| 8 | Ti-SiO$_2$ | 1-hex | 0.69 | 2.8 | 0.3 | 132.9 | 11.8 | 6.5 |
| 9 | Ti-ONa | 1-hex | 0.59 | 2.4 | 4.0 | 121.6 | 8.4 | 2.9 |
| 10 | Ti-ONa-SiO$_2$ | 1-hex | 1.38 | 5.5 | 2.5 | 127.6 | 56.9 | 3.2 |
| 11 | Ti | 4MP1 | 0.12 | 0.5 | 2.3 | 126.8 | 8.2 | 2.3 |
| 12 | Ti- SiO$_2$ | 4MP1 | 0.55 | 2.2 | 0.4 | 130.9 | 7.1 | 4.6 |
| 13 | Ti-ONa | 4MP1 | 0.52 | 2.1 | 2.8 | 126.2 | 11.5 | 2.1 |
| 14 | Ti-ONa-SiO$_2$ | 4MP1 | 1.87 | 7.5 | 1.3 | 128.2 | 79.4 | 3.0 |

[a]Conditions: The same metal loading (20 μmol/g) was used for all solid supports (10 μmol/g loading for **Ti-SiO$_2$**); Et$_2$AlCl: 500 eq., 3 mL (comonomer + n-heptane), $T = 30$ °C, Entry 1–6: Ti quantity = 0.25 μmol, ethylene pressure = 8 atm, time = 10 min; Entry 7-14: Ti quantity = 0.5 μmol, ethylene pressure = 2 atm, time = 30 min, comonomer: 2 mol·L$^{-1}$.
[b]Polymer yield and activity values are average of at least two runs. Activity = $10^6$ g mol$^{-1}$ h$^{-1}$.
[c]Comonomer incorporation ratio was determined by NMR in C$_2$D$_2$Cl$_4$ at 120 °C.
[d]Melting temperature determined by DSC.
[e]$M_n$: $10^4$ g mol$^{-1}$, $M_n$ and $M_w/M_n$ determined by GPC in trichlorobenzene at 150 °C.

polymerization, and copolymerization, subsequent studies focused on the practical aspects of this new supporting strategy, as well as new heterogeneous catalysts.

**Polymerization at high temperatures.** During the industrial production of polyethylene, high polymerization temperatures (70–110 °C) are typically required[47,48]. Extensive research efforts have been directed towards ligand modification to improve catalyst thermal stability[49]. Despite the effectiveness of this strategy, it requires complicated and costly syntheses. In this work, we demonstrate that the catalyst thermal stability can be greatly enhanced through heterogenization. Most importantly, all polymerization parameters (activity, polymer molecular weight, etc.) were improved while enhancing the thermal stability. For example, both **Ni** and **Ni-ONa** lost activity after 60 min at 100 °C during ethylene polymerization (Supplementary Fig. 8); however, both **Ni-ONa-SiO$_2$** and **Ni-ONa-MgO** remained highly active for 90 min at 100 °C, with **Ni-ONa-MgO** being most active and thermally stable. The differences are much more dramatic at higher polymerization temperatures. At 150 °C or 170 °C, **Ni** barely showed any activity, while both **Ni-ONa-MgO** and **Ni-F-ONa-MgO** remained active within 60 min (Fig. 3a, b). Reaction kinetic profile (real-time ethylene consumption detection curve for polymerization) agrees very well with the time-dependence polymerization studies (Fig. 3c). Similarly, **Ti-ONa-SiO$_2$** was both more active and more thermally stable than **Ti-ONa** and **Ti** (Fig. 3d).

Both **Ni-ONa-MgO** and **Ni-F-ONa-MgO** demonstrated surprisingly good thermal stability. Very high activities and high polymer molecular weights were maintained during ethylene homopolymerization even at 120 °C or 150 °C (Table 4, entries 1–3). Interestingly, copolymerization of tert-butyl acrylate at higher temperatures (100–140 °C) led to slightly higher activities and slightly lower copolymer molecular weights, along with a significantly higher comonomer incorporation (Table 4, entries 4–6). When the temperature was increased from 100 to 140 °C during copolymerization of ethylene with methyl 10-undecenoate, the activities and copolymer molecular weights slightly decreased (Table 4, entries 7–9); however, the comonomer incorporation ratios increased, which is similar to the situation observed for the copolymerization of ethylene with tert-butyl acrylate.

**Morphology control and large-scale polymerization.** The heterogeneous catalyst **Ni-ONa-SiO$_2$** generated free-flowing particles that did not stick to the surface of the reactor (Fig. 4a–c). In contrast, homogeneous catalyst **Ni** generated continuous polymer products that stuck to the polymerization reactor and stirrer (Fig. 4e–g). This is crucial to avoiding reactor fouling and achieving a continuous process, which is the industrially preferred strategy for polyethylene production.

Large-scale polymerization is usually difficult to control in academic labs due to thermal and mass transport issues, as well as the presence of large amounts of impurities from solvents and ethylene gas. **Ni-ONa-MgO** demonstrated great performances in a 2.5 L polymerization reactor (Table 4, entries 10 and 11, Fig. 4d, h). Specifically, at 20 atm ethylene pressure and 120 °C, an activity of $1.8 \times 10^8$ g mol$^{-1}$ h$^{-1}$ (150 g of polyethylene was obtained in a single run using 5 μmol of catalyst) and molecular weight of $1.62 \times 10^6$ g mol$^{-1}$ were achieved by using **Ni-ONa-MgO**. **Ni-F-ONa-MgO** showed higher activity and higher polyethylene molecular weight than **Ni-ONa-MgO** at 150 °C (Table 4, entry 12). The copolymerization of tert-butyl acrylate at 20 atm ethylene pressure and 120 °C using 30 μmol of **Ni-ONa-MgO** in a 2.5 L reactor generated 39.0 g of copolymer product with a molecular weight of $2.10 \times 10^5$ g mol$^{-1}$ (Table 4, entry 13). The copolymerization of methyl 10-undecenoate at 20 atm ethylene pressure and 120 °C using 30 μmol **Ni-ONa-MgO** in a 2.5 L reactor generated 47.5 g of copolymer product with a molecular weight of $4.56 \times 10^5$ g mol$^{-1}$ (Table 4, entry 14).

**Direct generation of polyethylene composites.** In the polyolefin industry, most polymers are reinforced with fibers or fillers to improve the balance between various material properties such as strength, stiffness, toughness, thermal/electrical conductivity, and flame retardancy[50,51]; however, one of the greatest obstacles to obtaining polyolefin composites is the chemical incompatibility between the hydrophobic polyolefin matrices and hydrophilic additives[52,53]. Numerous strategies have been developed to address this issue with varying success. One of the most direct and economic strategies is the immobilization of precatalysts on the surface of fibers or fillers, followed by the addition of aluminum cocatalysts and the in situ generation of composites through polymerization[54,55]; however, the addition of aluminum cocatalysts and their removal after polymerization may interfere

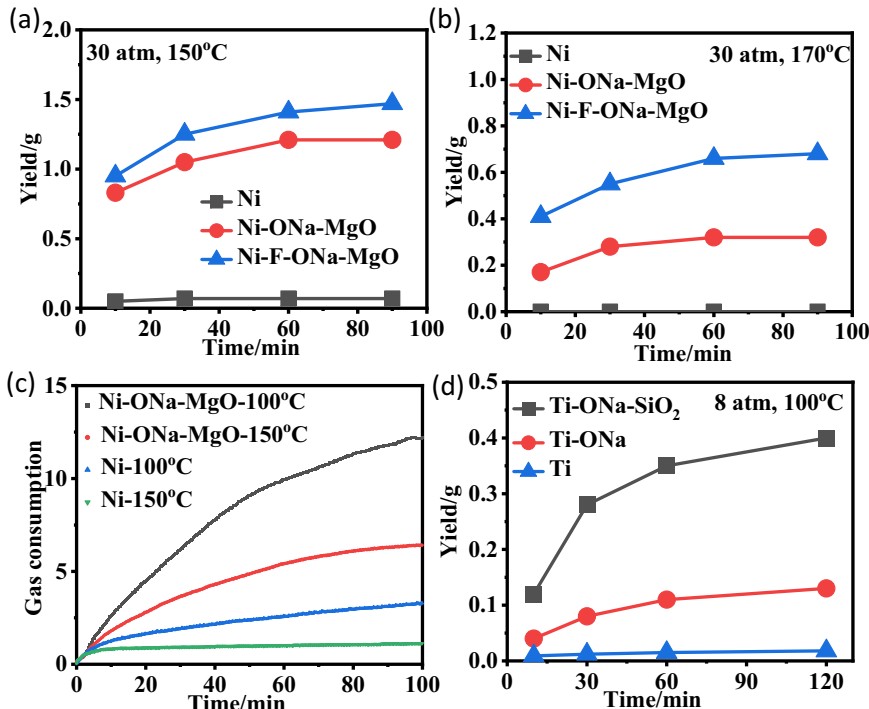

**Fig. 3 Polymerization at high temperatures. a**, **b** Time-dependence studies (yield versus time) of the nickel catalysts at 150 °C (0.1 μmol) and 170 °C (0.25 μmol), 30 atm. **c** Real-time ethylene consumption detection curve for polymerization of the nickel catalysts (0.1 μmol) at 150 °C and 100 °C, 30 atm. **d** Time-dependence studies (yield versus time) of the titanium catalysts (0.5 μmol) at 100 °C, 8 atm.

### Table 4 Ethylene polymerization and copolymerization at high temperatures and large scale[a].

| Ent. | Cat. | Comon. | [M]/ mol L⁻¹ | T/°C | Yield[b] /g | Act.[b] (10⁶) | Incorp.[c] mol% | $T_m$[d] /°C | $M_n$[e] (10⁴) | $M_w/M_n$[e] |
|---|---|---|---|---|---|---|---|---|---|---|
| 1 | Ni-ONa-MgO | – | – | 120 | 1.35 | 81.0 | – | 138.6 | 208.5 | 2.8 |
| 2 | Ni-ONa-MgO | – | – | 150 | 0.83 | 49.8 | – | 136.9 | 133.3 | 3.4 |
| 3 | Ni-F-ONa-MgO | – | – | 150 | 0.95 | 57.0 | – | 136.0 | 179.1 | 3.9 |
| 4 | Ni-ONa-MgO | | 0.2 | 100 | 0.29 | 0.1 | 2.0 | 125.6 | 4.0 | 2.4 |
| 5 | Ni-ONa-MgO | | 0.2 | 120 | 0.33 | 0.1 | 7.4 | 123.1 | 3.5 | 2.5 |
| 6 | Ni-ONa-MgO | | 0.2 | 140 | 0.39 | 0.2 | 6.6 | 120.9 | 3.2 | 2.3 |
| 7 | Ni-ONa-MgO | | 0.5 | 100 | 1.50 | 3.0 | 1.1 | 128.0 | 11.3 | 2.4 |
| 8 | Ni-ONa-MgO | | 0.5 | 120 | 1.38 | 2.8 | 1.8 | 127.6 | 7.8 | 2.9 |
| 9 | Ni-ONa-MgO | | 0.5 | 140 | 0.91 | 1.8 | 2.4 | 126.7 | 4.7 | 2.2 |
| 10 [f] | Ni-ONa-MgO | – | – | 120 | 150.00 | 180.0 | – | 137.8 | 161.8 | 2.5 |
| 11 [f] | Ni-ONa-MgO | – | – | 150 | 118.00 | 141.6 | – | 136.2 | 99.7 | 3.3 |
| 12 [f] | Ni-F-ONa-MgO | – | – | 150 | 125.00 | 150.0 | – | 136.2 | 154.1 | 4.2 |
| 13 [g] | Ni-ONa-MgO | | 0.2 | 120 | 39.00 | 1.3 | 1.3 | 129.5 | 21.0 | 2.9 |
| 14 [g] | Ni-ONa-MgO | | 0.5 | 120 | 47.50 | 3.2 | 0.4 | 132.4 | 45.6 | 2.9 |

[a]Conditions: 5 mL *n*-heptane; Entries 1–3: Ni quantity 0.1 μmol, time = 10 min, ethylene pressure = 30 atm; Entries 4–6: Ni quantity 5 μmol, time = 30 min, ethylene pressure = 8 atm; Entries 7–9: Ni quantity 1 μmol, *t* = 30 min, ethylene pressure = 8 atm.
[b]Polymer yield and activity values are average of at least two runs. Activity = 10⁶ g mol⁻¹ h⁻¹.
[c]Comonomer incorporation ratio was determined by ¹H NMR in $C_2D_2Cl_4$ at 120 °C.
[d]Melting temperature determined by DSC.
[e]$M_n$: 10⁴ g mol⁻¹, $M_n$ and $M_w/M_n$ determined by GPC in trichlorobenzene at 150 °C.
[f]Ni quantity 5 μmol, 1000 mL *n*-heptane in a 2.5 L polymerization reactor, time = 10 min, ethylene pressure = 20 atm.
[g]Ni quantity 30 μmol, 1000 mL *n*-heptane in a 2.5 L polymerization reactor, ethylene pressure = 20 atm; Entry 13: time = 60 min, Entry 14: time = 30 min.

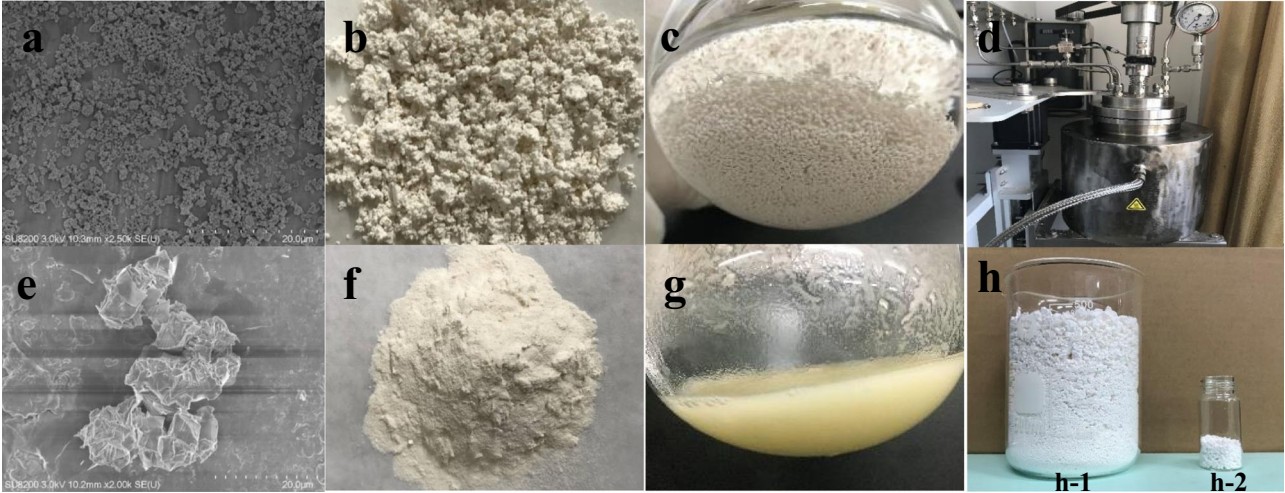

**Fig. 4 Polymer morphology and large-scale polymerization. a** SEM image of polyethylene samples prepared from **Ni-Na-SiO₂**. **b, c** Polyethylene samples prepared from **Ni-Na-SiO₂**. **d** 2.5 L polymerization reactor. **e** SEM image of polyethylene samples prepared from **Ni**. **f, g** Polyethylene samples prepared from **Ni**. **h-1** Polyethylene prepared in a 2.5 L polymerization reactor. **h-2** Polyethylene prepared in a 10 mL polymerization reactor.

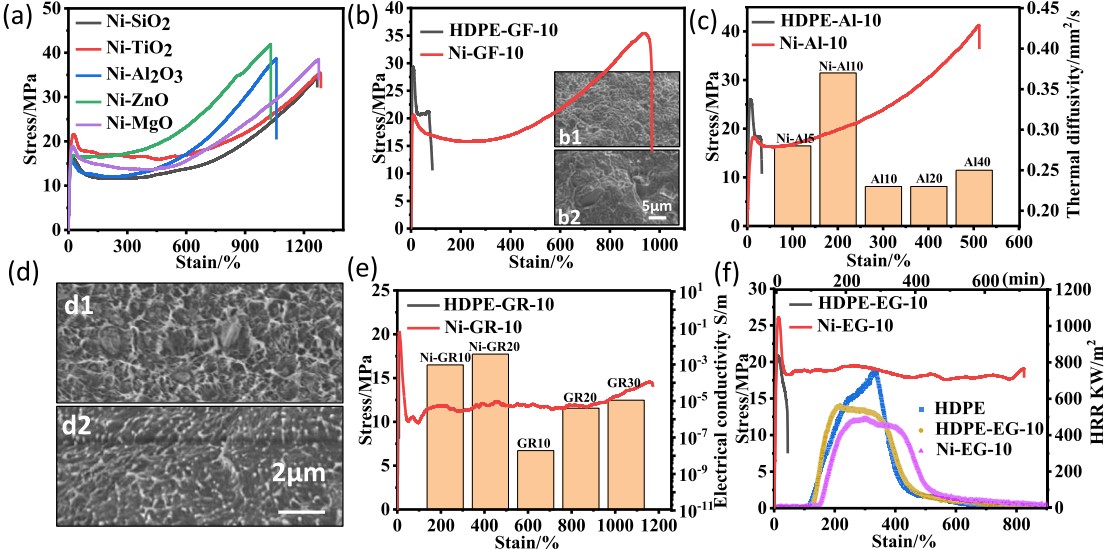

**Fig. 5 Properties of polyethylene composites.** The number in the sample name represents the approximate content of inorganic filler; the sample names containing "Ni" were prepared from in situ polymerization using supported catalysts, and the other samples were prepared from extrusion blending; GF, Al, GR and EG represent Glass fiber, Al₂O₃, Graphene and Expanded graphite, respectively. **a** Stress-strain curves of samples obtained using Ni-SiO₂ (Table 1, Entry 6), Ni-TiO₂ (Table 1, Entry 7), Ni-Al₂O₃ (Table 1, Entry 8), Ni-ZnO (Table 1, Entry 9), and Ni-MgO (Table 1, Entry 10). **b** Stress-strain curves of HDPE-GF-10 and Ni-GF-10. Inset: SEM images (b1 and b2) of Ni-GF-10 and HDPE-GF-10. **c** Stress-strain curves of HDPE-Al-10 and Ni-Al-10. The thermal diffusivity of Ni-Al5, Ni-Al10, Al10, Al20, and Al30. **d** SEM images (d1 and d2) of Al20 and Ni-Al20. **e** Stress-strain curves of HDPE-GR-10 and Ni-GR-10. Inset: the electrical conductivity of Ni-GR10, Ni-GR20, GR10, GR20, and GR30. **f** Stress-strain curves of HDPE-EG-10 and Ni-EG-10. Inset: the heat release rates of HDPE, HDPE-EG-10, and Ni-EG-10.

with interactions between the polymer matrices and fibers or fillers.

The **Ni-ONa** catalyst tethered on various inorganic supports does not require the addition of a cocatalyst or scavengers, making it a promising candidate for the generation of high-performance polyolefin composites. Indeed, the polymer samples bearing 1–2 wt% of inorganic supports (Table 1, entries 6–10) all showed tensile strengths close to 40 MPa and strain at break values well above 1000% (Fig. 5a). By carefully choosing the catalyst loading and polymerization conditions (temperature, time, ethylene pressure), it is very easy to obtain polyolefin composites with different filler

contents (Supplementary Table 5). Using 10 wt% glass fiber, the in situ generated composite showed a tensile strength of 35.5 MPa and a strain at a break value of 970% (Fig. 5b). In contrast, compounding polyethylene with 10 wt% glass fiber using a twin-screw extruder formed very brittle materials with a strain at break < 50%. Moreover, the in situ generated composite showed a homogeneous distribution of individual components in the SEM image (Fig. 5b1), while the sample obtained from extrusion showed clear phase separation (Fig. 5b2).

For Al₂O₃, the in situ generated composite showed a tensile strength of 40.2 MPa and a strain at break of 520% at 10 wt% filler

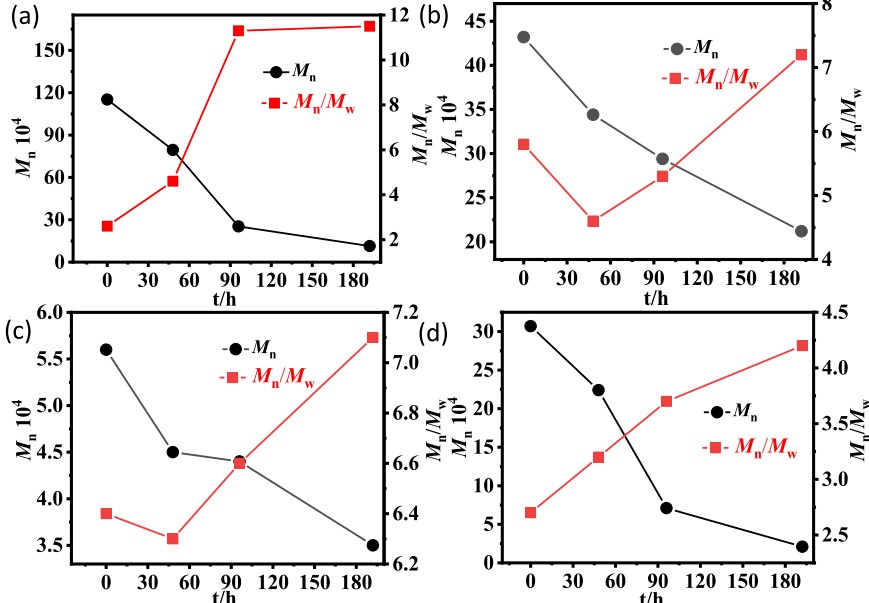

**Fig. 6 Photocatalytic degradation of polyethylene-TiO₂ composites. a** The in situ generated sample obtained using **Ni-ONa-TiO₂**. **b** The sample obtained by blending 95% high-molecular-weight polyethylene with 5 wt% TiO₂. **c** The sample obtained by blending 95% commercial polyethylene with 5 wt% TiO₂. **d** Sample prepared from the copolymerization of ethylene and methyl 10-undecylenate using **Ni-ONa-TiO₂**; polar monomer incorporation of 0.6%.

loading (Fig. 5c). In contrast, the compounded composite showed a strain at break < 50%. Even with 20 wt% Al₂O₃, the in situ generated composite showed a homogeneous distribution, while the sample obtained from extrusion showed clear phase separation (Fig. 5d). Similarly, the in situ generated polyethylene/graphene and polyethylene/expanded graphite composites all showed significantly better mechanical properties than their compounded counterparts (Fig. 5e, f).

More importantly, the homogeneous distribution of inorganic fillers in the polyolefin matrix makes it much easier to improve many important material properties. For example, the in situ generated polyethylene/Al₂O₃ composites showed great thermal conductivity (inset picture in Fig. 5c; thermal diffusivity: 0.28/0.38 for 5 wt%/10 wt% of Al₂O₃). In contrast, the polyethylene/Al₂O₃ composites obtained from extrusion showed much worse thermal conductivity even at much higher Al₂O₃ loadings (thermal diffusivity: 0.23/0.27 for 20 wt%/40 wt% of Al₂O₃). More strikingly, the electrical conductivities of the in situ generated polyethylene/graphene composites were three orders of magnitude higher than their extruded counterparts at similar graphene loadings (inset picture in Fig. 5e; $9.5 \times 10^{-4}$ versus $1.9 \times 10^{-8}$ S/m at 10 wt% graphene; $3.7 \times 10^{-3}$ versus $4.0 \times 10^{-6}$ S/m at 20 wt% graphene). Cone calorimetry is a useful lab-scale tool for evaluating the flammability of polymer composites. All parameters (time to ignition (TTI), time to peak heat release rate, peak heat release rate (PHRR), fire growth index (FGI)) indicated that the in situ generated polyethylene/expanded graphite possessed better flame retardancy than its extruded counterpart (inset picture in Fig. 5f, Supplementary Table 6).

**Photodegradation of polyethylene/TiO₂ composites**. The huge annual production of polyolefins and their superior strength and durability are profound end-of-life drawbacks, and their disposal has given rise to serious environmental concerns[56,57]. A number of solutions are currently being explored[58], and photocatalytic degradation in the presence of photocatalysts such as TiO₂ has emerged as a promising strategy[59]. The photodegradation of polyolefins to small-molecular-weight chains can trigger their subsequent biological degradation[60,61]; however, it is challenging

to prepare uniform composites of nonpolar polyolefins and polar photocatalyst nanoparticles, which have similar obstacles as the above-mentioned synthesis of various polyolefin composites. This severely limits the photodegradation efficiency and material properties of the targeted polyolefin composites. The addition of a compatibilizer or surface modification of the photocatalyst is required for the synthesis of uniform composites[62,63]. Our new heterogenization strategy makes it easy to prepare polyolefin composites with uniformly distributed photocatalysts with great mechanical properties.

Three polyethylene composite samples with 5 wt% TiO₂ were prepared from in situ polymerization and by blending a high-molecular-weight polyethylene sample and a low-molecular-weight commercial HDPE sample. The in situ generated sample showed much better mechanical properties than the compounded samples (35.9 MPa, 850% versus 28.0 MPa, 360%). After UV irradiation for 192 h, the molecular weight of the in situ generated sample dramatically decreased from $115.2 \times 10^4$ g mol⁻¹ to $11.4 \times 10^4$ g mol⁻¹, along with a significantly higher PDI value (Fig. 6a). In contrast, the molecular weights of the blended samples only decreased from $43.2 \times 10^4$ g mol⁻¹ to $21.2 \times 10^4$ g mol⁻¹, and $5.6 \times 10^4$ g mol⁻¹ to $3.5 \times 10^4$ g mol⁻¹, respectively (Fig. 6b, c). Furthermore, a copolymer was prepared from the copolymerization of ethylene and methyl 10-undecylenate using **Ni-ONa-TiO₂** with ca. 5 wt% of TiO₂ and 0.6 mol% comonomer. After UV irradiation for 192 h, the molecular weight of this copolymer sample decreased from $30.7 \times 10^4$ g mol⁻¹ to $2.1 \times 10^4$ g mol⁻¹, along with a greatly increased PDI value (Fig. 6d). Clearly, the incorporated polar-functional groups increased the efficiency of the photocatalytic degradation process. The presence of methine carbons along the polymer backbone resulted from enchaining long-spaced alkenoate are easier to oxidize during the degradation cascade. In addition, the absorption of ultraviolet light by polar groups may also contribute to this process. We are currently exploring the introduction of photosensitizing functional groups into the backbone of polyolefins through copolymerization to further enhance their photoresponsiveness.

In summary, we have developed a simple and general strategy for the heterogenization of various transition metal catalysts. By

installing an ionic tag, various transition metal catalysts (phosphinophenolate nickel, phenoxy-imine titanium) could be easily heterogenized on different solid supports. The resulting heterogeneous catalysts showed greatly enhanced properties (activity, stability, polymer molecular weight) during ethylene polymerization than their unsupported counterparts. More interestingly, the heterogeneous systems could incorporate more comonomers during copolymerization than the unsupported systems, which may originate from the interactions of the ionic tag with the surface of different solid supports. This strategy also provides great control over the polymer morphology, and can be used for large-scale polymerization at high temperatures. Furthermore, various polyolefin composites could be directly obtained by carefully choosing the solid support and polymerization conditions. This in situ synthetic procedure enables the dispersion of different fillers in the polyolefin matrix, which translates into great material properties (mechanical, thermal diffusivity, electrical conductivity, flame retardancy, etc.) even at low composite loadings. The utilization of a photoresponsive support (such as $TiO_2$) led to the generation of polyolefin composites with tunable photodegradation without sacrificing their mechanical properties, which may help to address the growing concerns over polyolefin disposal.

## Methods

**General methods and materials**. All experiments were carried out under a dry nitrogen atmosphere using standard Schlenk techniques or in a glovebox. The [1]H MAS NMR experiments were performed on a 14.1T Bruker Avance NEO 600 spectrometer using a 3.2 mm triple resonances MAS probe. Deuterated solvents used for Liquid NMR spectroscopy were dried and distilled prior to use. Liquid NMR spectra were recorded on a JNM-ECZR/S1 spectrometer at ambient temperature unless otherwise stated. The chemical shifts of the [1]H and [13]C NMR spectra were referenced to tetramethylsilane. Coupling constants are in Hz. Molecular weight and molecular weight distribution of the polymer were determined by gel permeation chromatography (GPC) with a PL-220 equipped with two Agilent PLgel Olexis columns at 150 °C using 1,2,4-trichlorobenzene as a solvent, and the calibration was made using polystyrene standard and are corrected for linear polyethylene by universal calibration using the Mark–Houwink parameters of Rudin: $K = 1.75 \times 10^{-2}$ cm$^3$/g and $R = 0.67$ for polystyrene and $K = 5.90 \times 10^{-2}$ cm$^3$/g and $R = 0.69$ for polyethylene. Dichloromethane, THF, and hexanes were purified in solvent purification systems. DSC measurements were performed on a TA Instruments DSC Q10. Samples (ca. 5 mg) were annealed by heating to 150 °C at 10 °C/min, cooled to 40 °C at 10 °C /min, and then analyzed while being heated to 150 °C at 10 °C/min. UV−vis absorption spectra were obtained using a Shimadzu 3600 spectrophotometer. Diethylaluminum chloride (1.0 mol/L in heptane) was purchased from Energy Chemistry and used as received. Powders of $SiO_2$ (~20 nm), $TiO_2$ (20–40 nm), $Al_2O_3$ (10–15 nm), ZnO (30–80 nm), MgO (~20 nm) were obtained from Nanjing XFNANO Materials Tech Co., Ltd., China. These Powders were treated in a tube furnace at 600 °C for 6 h before use. Expanded graphite and Graphene (Diameter: 5–10 μm, Thickness: 3–10 nm) were obtained from Nanjing XFNANO Materials Tech Co., Ltd., China. Ammonium polyphosphate ($n > 1000$) was purchased from Energy Chemistry. Glass fiber (8000 mesh) was purchased from Fuhua Nano New Materials Company. Expanded graphite, Graphene, Glass fiber and Ammonium polyphosphate were all treated with a certain amount of $Et_2AlCl$ (20 times equivalent of catalyst) before use.

**Mechanical properties**. Stress/strain experiments were performed at room temperature at 10 mm/min using a UTM2502 universal tester. At least three specimens of each polymer were tested. The test specimens had the following dimensions: gauge length, 28 mm; width, 2 mm; and thickness, 1 mm.

**Cone calorimeter**. The flammability test was performed on the cone calorimeter (FTT, UK) test according to ISO 5660 standard procedures, with the sample dimensions of $100 \times 100 \times 3$ mm$^3$. Each specimen was wrapped in aluminum foil and exposed horizontally to 35 kW/m$^2$ external heat flux.

**Scanning electron microscopy (SEM)**. The images of fracture surface for composites were obtained using a Hitachi Model X650 SEM system.

**Thermal diffusivity**. This was measured by Laser Thermal Instrument (Netzsch LFA467) at 30 °C.

**Electrical conductivity**. The 0.3 mm thick polymer film was placed between two custom-made copper sheets, and the resistance was measured three times with the Digital Multimeter (DELIXI ElECTRIC), and the average value was taken.

**Photodegradation experiments**. The decomposition of polymer films was investigated by exposing the samples to UV light under a UV exposure unit (300–400 nm, Irradiance=40 w/m$^2$, Test chamber temperature=35 °C).

**Procedure for polymerization**. In a typical experiment, a Biotage Endeavor Parallel Pressure Reactor with 8 built-in parallel high-pressure polymerization reactor each with a volume of 10 mL was used for ethylene polymerization. After adding a certain amount of catalyst and 5 or 3 mL solvent at the desired temperature, ethylene was inputted to start polymerization. At the end of the polymerization, the polymer product was filtered and dried at 45 °C for 24 h under vacuum. The real-time consumption curve of ethylene gas is directly monitored by this reactor. The consumption of ethylene gas is a relative value. The copolymer product was filtered, extracted with a Soxhlet extractor to remove the remaining comonomer, and dried at 45 °C for 24 h under vacuum. For the scale-up polymerization experiment, a high-pressure polymerization reactor with a volume of 2.5 L was used. The polymers used for morphology comparison in Fig. 4b, c, f, g were prepared using a 350 mL glass thick-walled pressure vessel.

## Data availability

All data necessary to support the conclusions of this paper are available in the supplementary materials, including materials, detailed experimental procedures, and characterization, as well as ESI-MS (Supplementary Figs. 25–29), NMR data (Supplementary Figs. 1, 2, 5, 10–24 and 30–60), DSC (Supplementary Figs. 61–130), GPC (Supplementary Figs. 131–214). All data can be supplied by the authors upon request.

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

## Acknowledgements

This work was supported by the National Key R&D Program of China (No. 2021YFA1501700), National Natural Science Foundation of China (No. 52025031, U19B6001, and U1904212), and K. C. Wong Education Foundation.

## Author contributions

C.Z. and G.S. contributed equally to the work. C.C. conceived the project. C.Z. and G.S. performed experiments regarding the synthesis and characterization. All authors contribute to data analysis and paper writing.

## Competing interests

The authors declare no competing interests.
