## [Peer Review File · Nature Communications]

A General Strategy for Heterogenizing Olefin Polymerization Catalysts and the Synthesis of Polyolefins and CompositesREVIEWER COMMENTS

Reviewer #1 (Remarks to the Author):

Chen and co-workers describe a method to generate heterogeneous olefin polymerization catalysts, one of which is tolerant to polar monomers. This is an emerging area that bridges academic and industrial labs, and I believe this paper represents a leap forward in understanding how to heterogenize late transition metal catalysts, which are harder to anchor to supports using “simple” methodologies. Chen and co-workers show that including an alkoxide on the ligand results in strong adsorption to various oxide surfaces. The ingenuity of this result lies in how easy it is to prepare these catalysts, as shown for a Ni, Ti, and Fe catalyst. When reading this I was reminded of the strong electrostatic adsorption methods that are very popular in the heterogeneous catalysis community, but to my knowledge have not been reported in the academic literature for olefin polymerization catalysts. The novelty and favorable properties of these catalysts supported on these oxides certainly makes this appropriate for a high impact publication, such as Nat Comm. Though I am very excited about the results, and strongly support publication, there are several revisions that need to be addressed prior to accepting the manuscript.

Introduction: There were several points that are either incorrect or poorly worded.

For example, “Despite the huge annual production of polyolefin materials, one of their biggest disadvantages is their non-polar nature, which significantly limits their applications in many areas.”⁶

Yes, this is technically true in a limited sense, but these non-polar materials are so widely available and tunable that the authors should consider revising this sentence to reflect the overall impact of these materials as opposed to limits in their applications in undefined areas.

“Academic polyolefin research has mainly focused on homogeneous systems due to their intrinsic advantages such as their defined molecular structures and rational modification, which make them useful for mechanistic studies.”

This is not true. The surface organometallic community originated in studies by Ballard and Yermakov to understand the exceptional activity of Ziegler-Natta compositions relative to organometallics known at the time (1970s). The field has continued to grow from these understandings, and relationships between these studies and “actual” catalysts for olefin polymerization represent some of the highest impact results from the field. The authors should revise the introduction to reflect these important contributions to the field.

Scheme 1 and the text describing Scheme 1 (lines 58-74) needs thorough revision. These revisions will also affect some of the conclusions made in the text in the results and discussion. My issue is that “face-up” or “face-down” has no structural basis in the design of a heterogeneous polymerization catalyst. For example, recent reports showing how an industrial metallocene/ $\text{AlR}_3/\text{Al}_2\text{O}_3$ catalyst self-assembles to form very active sites (ACS Cent Sci, 2021, 7, 1225) that are orders of magnitude more active than the catalysts described in this paper. The structure of the active site would be considered “face-down” by Scheme 1 because the metallocene fragment coordinates to surface sites, but these surface sites act as weakly coordinating anions. Therefore, the problem isn't “face-up” or “face-down,” but rather how the active site interacts with the support.

Related to this point, the authors do not have any evidence to show that Ni, Ti, nor Fe are actually “face-up” on these oxides. For Ti and Fe any studies about the nature of the metal support interaction are going to be further complicated by the structures of Et₂AlCl functionalized oxides (see: *J. Catal.* **2014**, *313*, 46-54.) or R₃Al functionalized oxides (see above ACS Cent Sci, and cited references within that paper for examples). Proving the conformations of supported species on a surface is very challenging, and will require extensive solid-state NMR spectroscopy and DFT modelling (see: *J. Am. Chem. Soc.* **2017**, *139*, 849 – 855). To be clear, these studies are, in my opinion, beyond the scope of this paper because of the elegance of the reactions that form the supported species and their overall utility. But I do hope these comments encourage the authors to seriously revise the text and Scheme 1.

Results and Discussion: These comments are rather minor compared to those listed above regarding points in the introduction.

How was metal loading in Figure S1 quantified? What are the metal loadings on the other supports? The latter point is important because metal loading is needed to accurately determine activities in Table 1.

Something is not right in Table 1, why are the polymer yields (in g) so much lower than expected for supported catalysts based on these activities?

Ni-ONa is certainly not soluble in heptane, is this also a heterogeneous catalyst?

Figure S2 has a typo. PMe₃ was used in the main text. The authors should also cite *Inorg. Chem.* 2021, 60, 10, 6946–6949; this study also showed high active site loading in a supported Ni-catalyst.

“FT-IR analysis showed that the C-O stretching peak shifted slightly after being supported (Figure S4). These results suggest that the interaction between the more basic and correspondingly more electron-donating solid support with the ONa moiety increased the electronic density of the nickel center.”

More evidence is needed to support this claim. FTIR measurements of adsorbed pyridine are often used in the heterogeneous community to assess Lewis acidity. The Ni catalysts should contain bound pyridine, and I would expect a shift in the sp² C=N stretch to support this claim. The authors could also consider contacting the materials with CO, which should form the acyl-CO complex. Adsorbed CO is another probe to assess electronics at the metal.

“During the copolymerization of ethylene/1-hexene(1-hex), Ti-SiO₂ showed more than an 8-fold decrease in comonomer incorporation than Ti (Table 3, entry 7 versus 6). The “face down” structural configuration in Ti-SiO₂ may significantly limit comonomer access to the metal center.”

This sentence should be revised to reflect changes made in the introduction.

Figure 1 has a typo on the blue triangles.

“Large-scale polymerization is usually difficult to control due to thermal and mass transport issues, as well as the presence of large amounts of impurities from solvents and ethylene gas.”

This is poorly worded. Industry makes over 100 million tons of polyolefin per year, so there is not a huge issue with mass transport, maintaining thermal control during polymerization, or removing solvents/cocatalysts

Reviewer #2 (Remarks to the Author):

A series of arenol-containing Ni, Ti, and Fe polymerization catalysts were supported on silica or related inorganic particles and applied to ethylene polymerization or copolymerization of ethylene with α -olefin or polar alkenes. Heterogenization was accomplished from the conjugate base of the ligand arenol group in each case. Comparisons were drawn between the supported catalysts and non-supported analogues with aryloxide, arenol, or ligands lacking the hydroxyl functional group altogether, which provided some noticeable differences in performance. For instance, cases of increased PE molecular weight and beneficial changes in catalyst activity can be observed, which is not uncommon when comparing supported vs. non-supported catalyst congeners. Applications of the support concept to incorporate fillers into the as-formed PE particles, or to embed titania nanoparticles were used to demonstrate a broader utility for these catalysts. A key weakness of this study is that it is conceptually incremental. It is rather common to leverage a reactive alcohol or arenol functional group at a distal site of a transition metal complex as a means to anchor the catalyst to a support; the criticism that alkylaluminum cocatalysts are prohibitive as a reagent in the anchoring protocol is largely moot considering the widespread use of such cocatalysts in industrial polymerizations. While the data presented in the study are extensive, the limited novelty is likely a prohibitive issue for this journal. With consideration of comments below, it should be more than suitable for a polymer-specific journal.

Additional comments:

1. Transition metal complexes have been routinely anchored to supports through protic functional groups, such as alcohols, carboxylic acids, and amines. Even if limited to the area of late transition metal-catalyzed insertion polymerization, it has solid precedent. Some selected examples include: Brookhart *Macromolecules* 2002, 35, 6074; *Macromolecules* 2006, 39, 6341; Sun *Organometallics* 2018, 37, 4002; Bernardo-Gusmao *New J. Chem.* 2017, 41, 2333; Shiono *Macromol. Chem. Phys.* 2014, 215, 1792.

2. The authors criticize the functional polyolefin molecular weights of prior work in heterogenized late metal polymerization catalysts. Prior work mentioned: Mn 16-24 kDa; this work Mn 3-15 (Table 2, entries 1-7). If the assertion here is the prior art is inadequate, then in fairness the same criticism should be levied against the present work. A related point (line 196): the assertion of a substantive molecular weight gain for E/MA copolymers using supported or homogeneous Ni catalysts is questionable based on the data in Table 2. All molecular weights fall into a low range for these examples irrespective of the homo-/heterogeneity of the catalyst.

3. The value of this study is fashioned around tolerance of a supported catalyst toward polar alkene monomers. However, two of the three catalysts reported have no activity toward copolymerizations of polar alkenes. Doesn't this undercut the stated purpose? Is there a reason three unique catalyst types each known for polar insertion copolymerization were not investigated?

4. The conclusion of thermal stability for Ni catalysts was not persuasive based on the data in Figure 1. There appears to be curvature in all data sets in parts a) and b) and the variations in the slope of the individual cases are nuanced. One could argue the slope difference may only reflect fluctuations in the intrinsic turnover frequency of the different catalysts with otherwise similar rates of thermal deactivation.

5. Line 199: Justification is needed for the claim that the Ni center is more electron rich in the

supported system. Wouldn't Ni-ONa necessarily be the most electron-rich in this series given it is associated with the most electropositive counterion (Na⁺)?

6. Line 421: It is not intuitive how a long-spaced ester enhances main chain scission. Alternatively, the presence of methine carbons along the polymer backbone, which result from enchaining long-spaced alkenoates, are much easier to oxidize during the degradation cascade.

7. Line 137: Unjustified claim: what data supports the claim hydroxyl groups can be easily installed onto a tert-butyl fragment?

8. Table 4 is described as examples of copolymerization, but it is unclear that is the case looking at the data and footnotes for this table.

Reviewer #3 (Remarks to the Author):

Report on the article: A General Strategy for the Heterogenization of Olefin 1 Polymerization Catalysts and the Synthesis of Polyolefins and Composites
By Chen Zou, Guifu Si, Changle Chen

This work describes a powerful strategy of anchoring organometallic precatalysts to an inorganic phase, affording highly efficient ethylene polymerization catalysts. Copolymerization with polar monomers is also achieved with such supported catalysts. The results in terms of polymer materials properties are outstanding when compared to more conventional way to support catalysts. To my opinion, this work needs little improvement, and then it will be worth to be published in Nature Communications, providing the points below are properly addressed.

One important concern is about the molecular structure of Ni-OH (scheme 2): I was wondering how the Ni-Me bond can coexist with the p-hydroxy moiety? This could be possibly addressed by determining the X-Ray structure of NiOH complex. Otherwise please discuss this somewhat unexpected feature.

On several occasions, the "face down" and "face up" structural configurations are discussed (e.g. in the discussion of Ti catalysts just after Table 2. For a better understanding, I recommend to represent these configurations by a schematic drawing.

In this paragraph relative to Ti catalysts, the comonomer incorporation is reported as "slightly decreasing" for TiONa-SiO₂ compared with TiONa. The term "slightly" is inappropriate (4.0 to 2.5%), and should be replaced by "significant".

Run 4 in table 3 looks very similar to run 5 in table 4: exactly the same values of yield/activity/T_m (1.62/38.9/137.5, respectively), but different Mn/Đ (289.7/2.0). Please check these data.

To efficiently present the thermal stability of catalysts, a figure that displays the kinetics of a given catalyst at several temperatures (e.g. Ni at 80 and 100 °C, or Ti at 30 and 100 °C) would be of interest.

p.14: Morphology control paragraph: specify Figure 2, efg and Figure 2, abc

Additional comments:

In Tables 1, 2, 3 (footnote): catalyst quantity (in μmol) should be replaced by metal quantity (in μmol)

In Tables 2, 3, 5: Incorp. instead of Incrop.

Table 3 title: ethylene polymerization and copolymerization...

Table 4 title: ethylene polymerization and not copolymerization...

P. 12, top: "...the above-mentioned studies on complexes synthesis..."

P. 12, mid: "The generality...makes it easy to support **a mix of** two or more precatalysts..."

In references 4, 5, 8, 10, 12, 16, 19, 41, 50 the names of all authors must be completed; reference 46 is not properly written

Experimental:

P. S3, 2.2 Catalysts preparation: line 2, Catalysts Ni; p. S4, line 9, Catalysts Ti; p. S4, line 19, Catalysts Fe

Page S5, line 5: "...nitrogen, **d**issolve..."

P. S6, 2.3 preparation of supported catalysts, line 7: 1 umol should be 1 μmol

Reviewer #1

Chen and co-workers describe a method to generate heterogeneous olefin polymerization catalysts, one of which is tolerant to polar monomers. This is an emerging area that bridges academic and industrial labs, and I believe this paper represents a leap forward in understanding how to heterogenize late transition metal catalysts, which are harder to anchor to supports using “simple” methodologies. Chen and co-workers show that including an alkoxide on the ligand results in strong adsorption to various oxide surfaces. The ingenuity of this result lies in how easy it is to prepare these catalysts, as shown for a Ni, Ti, and Fe catalyst. When reading this I was reminded of the strong electrostatic adsorption methods that are very popular in the heterogeneous catalysis community, but to my knowledge have not been reported in the academic literature for olefin polymerization catalysts. The novelty and favorable properties of these catalysts supported on these oxides certainly makes this appropriate for a high impact publication, such as Nat Comm. Though I am very excited about the results, and strongly support publication, there are several revisions that need to be addressed prior to accepting the manuscript.

Introduction: There were several points that are either incorrect or poorly worded.

For example, “Despite the huge annual production of polyolefin materials, one of their biggest disadvantages is their non-polar nature, which significantly limits their applications in many areas.”

Yes, this is technically true in a limited sense, but these non-polar materials are so widely available and tunable that the authors should consider revising this sentence to reflect the overall impact of these materials as opposed to limits in their applications in undefined areas.

Answer: We agree with the reviewer and have revise this sentence to: The introduction of polar-functional groups into the otherwise non-polar backbone of polyolefins could further broaden their applications in many areas.

“Academic polyolefin research has mainly focused on homogeneous systems due to their intrinsic advantages such as their defined molecular structures and rational modification, which make them useful for mechanistic studies.”

This is not true. The surface organometallic community originated in studies by Ballard and Yermakov to understand the exceptional activity of Ziegler-Natta compositions relative to organometallics known at the time (1970s). The field has continued to grow from these understandings, and relationships between these studies and “actual” catalysts for olefin polymerization represent some of the highest impact results from the field. The authors should revise the introduction to reflect these important contributions to the field.

Answer: Thanks a lot for your helpful comments. We have carefully revised this section to reflect these important contributions to the field.

In contrast, homogeneous systems possess the advantages of defined molecular structures and rational modification, which make them useful for mechanistic studies. The surface organometallic community originated in the 1970s to understand the exceptional activity of Ziegler-Natta compositions relative to organometallics known at the time.

Scheme 1 and the text describing Scheme 1 (lines 58-74) needs thorough revision. These revisions will also affect some of the conclusions made in the text in the results and discussion. My issue is that “face-up” or “face-down” has no structural basis in the design of a heterogeneous polymerization catalyst. For example, recent reports showing how an industrial metallocene/ $\text{AlR}_3/\text{Al}_2\text{O}_3$ catalyst self-assembles to form very active sites (ACS Cent Sci, 2021, 7, 1225) that are orders of magnitude more active than the catalysts described in this paper. The structure of the active site would be considered “face-down” by Scheme 1 because the metallocene fragment coordinates to surface sites, but these surface sites act as weakly coordinating anions. Therefore, the problem isn’t “face-up” or “face-down,” but rather how the active site interacts with the support.

Answer: We have carefully read the reviewer’s comments and revised this section accordingly.

We have cited this reference: ACS Cent Sci, 2021, 7, 1225

We have deleted the description of face-down or face-up in the whole manuscript. We have deleted the chemical structure of the “face-up” drawing in Scheme 1.

We have deleted this sentence in the original manuscript: Furthermore, the nature of these interactions usually affords a "face-down" structural configuration of the metal center, which significantly limits the accessibility of active sites to monomers and comonomers.

Related to this point, the authors do not have any evidence to show that Ni, Ti, nor Fe are actually “face-up” on these oxides. For Ti and Fe any studies about the nature of the metal support interaction are going to be further complicated by the structures of Et_2AlCl functionalized oxides (see: *J. Catal.* **2014**, 313, 46-54.) or R_3Al functionalized oxides (see above ACS Cent Sci, and cited references within that paper for examples). Proving the conformations of supported species on a surface is very challenging, and will require extensive solid-state NMR spectroscopy and DFT modelling (see: *J. Am. Chem. Soc.* **2017**, 139, 849 – 855). To be clear, these studies are, in my opinion, beyond the scope of this paper because of the elegance of the reactions that form the supported species and their overall utility. But I do hope these comments encourage the authors to seriously revise the text and Scheme 1.

Answer: Thank you very much for the helpful suggestions.

The nickel catalyst could interact with the surface in two ways: Ni center with surface (face down) or OM with surface (face up). Since narrow molecular weight distributions were observed for all the supported catalysts in ethylene polymerization or copolymerization, there could only be one type of interaction. We have shown that if we remove the OM ionic tag for either Ni or Ti catalysts, the interaction with the support is very weak (catalyst loading is much lower). And the copolymerization studies provide supporting evidence (increased comonomer incorporation after support) for the interaction of OM with surface in this system.

Although we believe the strong interaction between the ionic tag (OM in this case) with the surface should prevail in this system, we indeed lack direct evidence. Therefore, we have deleted the description of face-down or face-up in the whole manuscript.

Considering both Reviewer #1’s comments on Fe system due to the usage of aluminum cocatalyst and Reviewer #2’s comments on the lack of copolymerization studies on Fe system, we have decided to delete the works on Fe system in the revised manuscript.

To provide further evidence for the advantages of this heterogenization strategy, we included another nickel complex in the revised manuscript.

Inspired by Mecking and coworkers' recent works on perfluorophenyl-substituted phosphinophenolate nickel catalyst, the corresponding OH and ONa tagged nickel catalysts (**Ni-F-OH** and **Ni-F-ONa**) were also prepared. The polymerization and copolymerization properties of this new catalyst were better than the original catalyst **Ni-ONa**.

Results and Discussion: These comments are rather minor compared to those listed above regarding points in the introduction.

How was metal loading in Figure S1 quantified? What are the metal loadings on the other supports? The latter point is important because metal loading is needed to accurately determine activities in Table 1.

Answer: Thanks a lot for your comments. According to the reviewer's comments, we have added the detailed description in the supporting information.

The pretreated support (1 g) was stirred in 5 mL toluene. A 10 $\mu\text{mol/mL}$ catalyst toluene solution was added dropwise, until the supernatant liquid showed UV signal. The supported catalyst was filtered, rinsed with toluene three times until the filtrate showed no UV signal. The metal content was measured by ICP, and the maximum catalyst load was calculated accordingly.

The same metal loading (20 $\mu\text{mol/g}$) was used for all the other supports in Table 1. The current loading is lower than the maximum loading value for all of these supports. We have provided these information in the revised manuscript.

Something is not right in Table 1, why are the polymer yields (in g) so much lower than expected for supported catalysts based on these activities?

Answer: Thanks a lot for your comments. The supported catalysts are much more active, therefore smaller amount of catalysts was used (0.1 μmol versus 0.5 μmol for homogeneous catalyst). This leads to the differences. To avoid confusion, we have included a column for the amount of Ni (μmol) in Table 1.

Ni-ONa is certainly not soluble in heptane, is this also a heterogeneous catalyst?

Answer: Thanks a lot for your comments. This is a great point. Ni-ONa should be considered a heterogeneous catalyst. Therefore, we have revised our discussions in the manuscript to supported and non-supported catalysts.

Figure S2 has a typo. PMe_3 was used in the main text. The authors should also cite *Inorg. Chem.* 2021, 60, 10, 6946–6949; this study also showed high active site loading in a supported Ni-catalyst.

Answer: Thanks a lot for your comments. According to the reviewer's comments, we corrected the typos in the supporting information. We have also cited this study *Inorg. Chem.* 2021, 60, 6946-6949 in the manuscript.

“FT-IR analysis showed that the C-O stretching peak shifted slightly after being supported (Figure S4). These results suggest that the interaction between the more basic and correspondingly more electron-donating solid support with the ONa moiety increased the electronic density of the nickel center.”

More evidence is needed to support this claim. FTIR measurements of adsorbed pyridine are often used in the heterogeneous community to assess Lewis acidity. The Ni catalysts should contain bound pyridine, and I would expect a shift in the sp^2 C=N stretch to support this claim. The authors could also consider contacting the materials with CO, which should form the acyl-CO complex. Adsorbed CO is another probe to assess electronics at the metal.

Answer: Thanks a lot for your great suggestions. According to the reviewer’s comments, we have studied the FTIR spectrum of the supported nickel catalyst, in Figure 1, The C=N stretching peak of Ni-ONa-TiO₂, Ni-ONa-Al₂O₃, Ni-ONa-ZnO and Ni-ONa-MgO are at 1598.21 cm⁻¹, 1595.18 cm⁻¹, 1589.12 cm⁻¹ and 1588.73 cm⁻¹ respectively. The wavenumber of the C=N stretching peak decreases slightly with the increase of the alkalinity of the support, indicating that the electron density of the metal center increases with the alkalinity of the support.

We also used CO as the probe to assess electronics at the metal as suggested by the reviewer. However, we did not find the characteristic peak of CO in the FTIR spectrum after exposing the catalysts to CO gas.

Figure 1. IR spectra of Ni-based supported catalysts.

“During the copolymerization of ethylene/1-hexene(1-hex), Ti-SiO₂ showed more than an 8-fold decrease in comonomer incorporation than Ti (Table 3, entry 7 versus 6). The “face down” structural configuration in Ti-SiO₂ may significantly limit comonomer access to the metal center.”

This sentence should be revised to reflect changes made in the introduction.

Answer: Thanks a lot for your comments. We have revised the corresponding discussions.

Figure 1 has a typo on the blue triangles.

Answer: Thanks a lot for your comments. According to the reviewer’s comment, we corrected the typos in the Figure 1 in the manuscript.

“Large-scale polymerization is usually difficult to control due to thermal and mass transport issues, as well as the presence of large amounts of impurities from solvents and ethylene gas.”

This is poorly worded. Industry makes over 100 million tons of polyolefin per year, so there is not a huge issue with mass transport, maintaining thermal control during polymerization, or removing solvents/cocatalyst.

Answer: Thanks a lot for your comments. We agree with the reviewer. Polymerization at 2.5 liter scale is challenging for most academic labs, and there are very few literature reports for such scale olefin polymerization. As a matter of fact, many of our previously reported nickel catalysts showed much worse performance at 2.5 liter scale comparing with small scale (10~100 mL) polymerization. We have revised this sentence to: Large-scale polymerization is usually difficult to control in academic labs.

Reviewer #2

A series of arenol-containing Ni, Ti, and Fe polymerization catalysts were supported on silica or related inorganic particles and applied to ethylene polymerization or copolymerization of ethylene with α -olefin or polar alkenes. Heterogenization was accomplished from the conjugate base of the ligand arenol group in each case. Comparisons were drawn between the supported catalysts and non-supported analogues with aryloxide, arenol, or ligands lacking the hydroxyl functional group altogether, which provided some noticeable differences in performance. For instance, cases of increased PE molecular weight and beneficial changes in catalyst activity can be observed, which is not uncommon when comparing supported vs. non-supported catalyst congeners. Applications of the support concept to incorporate fillers into the as-formed PE particles, or to embed titania nanoparticles were used to demonstrate a broader utility for these catalysts. A key weakness of this study is that it is conceptually incremental. It is rather common to leverage a reactive alcohol or arenol functional group at a distal site of a transition metal complex as a means to anchor the catalyst to a support; the criticism that alkylaluminum cocatalysts are prohibitive as a reagent in the anchoring protocol is largely moot considering the widespread use of such cocatalysts in industrial polymerizations. While the data presented in the study are extensive, the limited novelty is likely a prohibitive issue for this journal. With consideration of comments below, it should be more than suitable for a polymer-specific journal.

Additional comments:

1. Transition metal complexes have been routinely anchored to supports through protic functional groups, such as alcohols, carboxylic acids, and amines. Even if limited to the area of late transition metal-catalyzed insertion polymerization, it has solid precedent. Some selected examples include: Brookhart *Macromolecules* 2002, 35, 6074; *Macromolecules* 2006, 39, 6341; Sun *Organometallics* 2018, 37, 4002; Bernardo-Gusmao *New J. Chem.* 2017, 41, 2333; Shiono *Macromol. Chem. Phys.* 2014, 215, 1792.

Answer: Thanks a lot for your comments. We have cited these and related references.

These references indeed studied the heterogenization of metal complexes bearing alcohols and amines. However, they require the usage of aluminum treated solid support. Otherwise, the interaction of the alcohols or amines with the solid surface is not strong enough to afford stable heterogeneous catalysts. Most importantly, none of these studies investigated copolymerization of ethylene with polar monomers.

Reviewer #1 also supported the novelty of our strategy: When reading this I was reminded of the strong electrostatic adsorption methods that are very popular in the heterogeneous catalysis community, but to my knowledge have not been reported in the academic literature for olefin polymerization catalysts.

As a matter of fact, we recently reported the generation of heterogeneous nickel catalyst through alcohol induced hydrogen bonding (*Nat. Commun.* **2020**, *11*, 372). The homogeneous nickel catalyst is highly active in ethylene copolymerization with a series of polar comonomers, while the heterogeneous counterpart showed no activity at all in copolymerization. These results clearly demonstrated the advantages of this heterogenization strategy in generating supported late transition metal catalysts, especially for the purpose of copolymerization reactions.

2. The authors criticize the functional polyolefin molecular weights of prior work in heterogenized late metal polymerization catalysts. Prior work mentioned: Mn 16-24 kDa; this work Mn 3-15 (Table 2, entries 1-7). If the assertion here is the prior art is inadequate, then in fairness the same criticism should be levied against the present work. A related point (line 196): the assertion of a substantive molecular weight gain for E/MA copolymers using supported or homogeneous Ni catalysts is questionable based on the data in Table 2. All molecular weights fall into a low range for these examples irrespective of the homo-/heterogeneity of the catalyst.

Answer: Thanks a lot for your very helpful comments.

To clear this point and provide a clear picture for the superior properties of our newly reported catalysts, we have included detailed comparison of the activity, polymer molecular weight and comonomer incorporation between literature reported homogeneous and heterogeneous catalysts and **Ni-ONa-MgO/Ni-F-ONa-MgO** (Figure S7 and Tables S1-S4). Clearly, our newly reported nickel catalysts are among the most potent catalysts in ethylene polymerization and copolymerization with polar comonomers.

Figure S7. (a) Comparison of activity and polymer molecular weight between representative

reported homogeneous nickel catalysts and **Ni-ONa-MgO/Ni-F-ONa-MgO**. (b) Comparison of activity and polymer molecular weight between representative reported heterogeneous nickel and palladium catalysts and **Ni-ONa-MgO/Ni-F-ONa-MgO**. (c) Comparison of polymer molecular weight and comonomer incorporation between representative reported homogeneous nickel/palladium catalysts and **Ni-ONa-MgO/Ni-F-ONa-MgO** in the copolymerization of ethylene with methyl acrylate or *tert*-butyl acrylate. (d) Comparison of polymer molecular weight and comonomer incorporation between representative reported heterogeneous nickel/palladium catalysts and **Ni-ONa-MgO/Ni-F-ONa-MgO** in the copolymerization of ethylene with methyl acrylate, *tert*-butyl acrylate or 10-methyl undecanoate.

To provide further evidence for the advantages of this heterogenization strategy, we included another nickel complex in the revised manuscript.

Inspired by Mecking and coworkers' recent works on perfluorophenyl-substituted phosphinophenolate nickel catalyst, the corresponding OH and ONa tagged nickel catalysts (**Ni-F-OH** and **Ni-F-ONa**) were also prepared. The polymerization and copolymerization properties of this new catalyst were better than the original catalyst **Ni-ONa**.

3. The value of this study is fashioned around tolerance of a supported catalyst toward polar alkene monomers. However, two of the three catalysts reported have no activity toward copolymerizations of polar alkenes. Doesn't this undercut the stated purpose? Is there a reason three unique catalyst types each known for polar insertion copolymerization were not investigated?

Answer: Thanks a lot for your comments. It is actually highly challenging to utilize Ti catalysts or Fe catalysts to copolymerize ethylene with polar comonomers. We have included the corresponding studies and comments in the revised manuscript for the Ti catalyzed ethylene copolymerization with polar comonomers:

It has been recognized as a great challenge to copolymerize ethylene with polar comonomers using titanium-based catalysts.^{47,48} Under the current conditions, the unsupported catalyst **Ti** showed no activity at all in the copolymerization of ethylene with 10-methyl undecanoate (Table 3, entry 15). Interestingly, the supported catalyst **Ti-ONa-SiO₂** showed polymerization activity but no comonomer incorporation (Table 3, entry 16). In contrast, **Ti-ONa-MgO** showed higher activity and polymer molecular weight than **Ti-ONa-SiO₂**, along with the capability of comonomer incorporation (Table 3, entry 17). These results may originate from the electron-donating ability of the ionic tag as well as its interaction with the surface of different solid supports.

We also performed copolymerization studies using homogeneous and heterogeneous Fe catalysts, which showed no activity at all. Considering both Reviewer #1's comments on Fe system due to the usage of aluminum cocatalyst and Reviewer #2's comments on the lack of copolymerization studies on Fe system, we have decided to delete the works on Fe system in the revised manuscript.

4. The conclusion of thermal stability for Ni catalysts was not persuasive based on the data in

Figure 1. There appears to be curvature in all data sets in parts a) and b) and the variations in the slope of the individual cases are nuanced. One could argue the slope difference may only reflect fluctuations in the intrinsic turnover frequency of the different catalysts with otherwise similar rates of thermal deactivation.

Answer: Thanks a lot for your helpful comments. At 100 °C, the differences are indeed not persuasive. To clarify this point, we have performed time dependence studies at 150 °C and 170 °C. The differences are much more dramatic at higher polymerization temperatures. At 150 °C or 170 °C, Ni barely showed any activity, while both Ni-ONa-MgO and Ni-F-ONa-MgO remained active within 60 min (Figures 1b and 1c). Reaction kinetic profile (real-time ethylene consumption detection curve for polymerization) agrees very well with the time dependence polymerization studies (Figure 1c).

Figure 1. (a) and (b) Time-dependence studies (yield versus time) of the nickel catalysts at 150 °C (0.1 μmol) and 170 °C (0.25 μmol), 30 atm. (c) Real-time ethylene consumption detection curve for polymerization. (d) Time-dependence studies (yield versus time) of the titanium catalysts (0.5 μmol) at 100 °C, 8 atm.

5. Line 199: Justification is needed for the claim that the Ni center is more electron rich in the supported system. Wouldn't Ni-ONa necessarily be the most electron-rich in this series given it is associated with the most electropositive counteraction (Na^+)?

Answer: Thanks a lot for your comments. Reviewer #1 raised the same question on this issue. In Figure S5, solid-state ¹H MAS NMR analysis of these heterogeneous catalysts showed that the resonances of the pyridine moiety gradually shifted downfield in the order: Ni-ONa-SiO₂ > Ni-ONa-Al₂O₃ > Ni-ONa-MgO. For Figure 6 (FT-IR), the C=N stretching peak of Ni-ONa-TiO₂, Ni-ONa-Al₂O₃, Ni-ONa-ZnO and Ni-ONa-MgO are at 1598.21 cm⁻¹, 1595.18 cm⁻¹, 1589.12 cm⁻¹ and 1588.73 cm⁻¹ respectively. The wavenumber of the C=N stretching peak decreases slightly with the increase of the alkalinity of the support. These results indicating that the electron density of the metal center increases with the alkalinity of the support.

Figure S5. ^1H MAS NMR spectrum of supported catalysts **Ni-ONa-SiO₂**, **Ni-ONa-Al₂O₃** and **Ni-ONa-MgO**.

Figure S6. IR spectra of Ni-based supported catalysts. The C=N stretching peak of **Ni-ONa-TiO₂**, **Ni-ONa-Al₂O₃**, **Ni-ONa-ZnO** and **Ni-ONa-MgO** are at 1598.21 cm^{-1} , 1595.18 cm^{-1} , 1589.12 cm^{-1} and 1588.73 cm^{-1} , respectively. The wavenumber of the C=N stretching peak

decreases slightly with the increase of the alkalinity of the support, indicating that the electron density of the metal center increases with the alkalinity of the support.

6. Line 421: It is not intuitive how a long-spaced ester enhances main chain scission. Alternatively, the presence of methine carbons along the polymer backbone, which result from enchaining long-spaced alkenoates, are much easier to oxidize during the degradation cascade.
Answer: Thanks a lot for your very helpful comments. What you described "the presence of methine carbons along the polymer backbone, which result from enchaining long-spaced alkenoates, are much easier to oxidize during the degradation cascade." is probably one of the important reasons. It may also be because the absorption of ultraviolet light by polar groups has a certain effect on the degradation of the polymer.

According to the reviewer's comments, we have revised the relevant statement in the manuscript:

Clearly, the incorporated polar functional groups increased the efficiency of the photocatalytic degradation process. The presence of methine carbons along the polymer backbone resulted from enchaining long-spaced alkenoate are easier to oxidize during the degradation cascade. In addition, the absorption of ultraviolet light by polar groups may also contribute to this process.

7. Line 137: Unjustified claim: what data supports the claim hydroxyl groups can be easily installed onto a tert-butyl fragment?

Answer: Thanks a lot for your comments. The original statement is: "For example, the hydroxyl group could be installed at the *P-Ph*, *P-Ar*, or *tBu* positions,"

As a matter of fact, we have already prepared nickel complex with a hydroxyl group at the position of ter-butyl group. Unfortunately, this catalysts showed much lower activity than the current catalysts.

To avoid controversy, we have removed this sentence according to the reviewer's comments.

8. Table 4 is described as examples of copolymerization, but it is unclear that is the case looking at the data and footnotes for this table.

Answer: Thanks a lot for your comments. As mentioned above, we have deleted the studies on Fe catalyst.

Reviewer #3

Report on the article: A General Strategy for the Heterogenization of Olefin 1 Polymerization Catalysts and the Synthesis of Polyolefins and Composites

By Chen Zou, Guifu Si, Changle Chen

This work describes a powerful strategy of anchoring organometallic precatalysts to an inorganic phase, affording highly efficient ethylene polymerization catalysts. Copolymerization with polar monomers is also achieved with such supported catalysts. The results in terms of polymer materials properties are outstanding when compared to more conventional way to support catalysts. To my opinion, this work needs little improvement, and then it will be worth to be published in Nature Communications, providing the points below are properly addressed.

One important concern is about the molecular structure of Ni-OH (scheme 2): I was wondering how the Ni-Me bond can coexist with the p-hydroxy moiety? This could be possibly addressed by determining the X-Ray structure of NiOH complex. Otherwise please discuss this somewhat unexpected feature.

Answer: Thanks a lot for your helpful comments. This is indeed an important issue.

^1H NMR characterization showed the coexistence of Ni-Me and Ar-OH signals in both **Ni-OH** and **Ni-F-OH**, and the integration of Ni-Me signal matched very well with the rest of the ligand signals (Figure S1). Furthermore, the addition of *p*-cresol to nickel complex without the OH group (**Ni**) indicated no reaction at all (Figure S2).

Figure S1. Coexistence of Ni-Me and Ar-OH signals in **Ni-OH** in ^1H NMR analysis.

Figure S2. ^1H NMR analysis showing no reaction between p-cresol and Ni.

On several occasions, the "face down" and "face up" structural configurations are discussed (e.g. in the discussion of Ti catalysts just after Table 2. For a better understanding, I recommend to represent these configurations by a schematic drawing.

Answer: Thanks a lot for your comments. In response to Reviewer #1's comments (see above), we have deleted the discussions on "face down" and "face up".

In this paragraph relative to Ti catalysts, the comonomer incorporation is reported as "slightly decreasing" for TiONa-SiO₂ compared with TiONa. The term "slightly" is inappropriate (4.0 to 2.5%), and should be replaced by "significant".

Answer: Thanks a lot for your helpful comments. To avoid confusion, we have included the comparison of incorporation values in the revised manuscript:

In contrast, the comonomer incorporation was decreased from 4.0 mol% for **Ti-ONa** to 2.5 mol% for **Ti-ONa-SiO₂** (Table 3, entry 10 versus 9).

Run 4 in table 3 looks very similar to run 5 in table 4: exactly the same values of yield/activity/T_m (1.62/38.9/137.5, respectively), but different Mn/Đ (289.7/2.0). Please check these data.

Answer: Thanks a lot for your comments. We have deleted the studies on Fe catalysts (Table 4).

To efficiently present the thermal stability of catalysts, a figure that displays the kinetics of a given catalyst at several temperatures (e.g. Ni at 80 and 100 °C, or Ti at 30 and 100 °C) would be of interest.

Answer: Thanks a lot for your helpful comments. We have performed time dependence studies at 150 °C or 170 °C. The differences are much more dramatic at higher polymerization

temperatures. At 150 °C or 170 °C, Ni barely showed any activity, while both Ni-ONa-MgO and Ni-F-ONa-MgO remained active within 60 min (Figures 1b and 1c). Reaction kinetic profile (real-time ethylene consumption detection curve for polymerization) agrees very well with the time dependence polymerization studies (Figure 1c).

Figure 1. (a) and (b) Time-dependence studies (yield versus time) of the nickel catalysts at 150 °C (0.1 μmol) and 170 °C (0.25 μmol), 30 atm. (c) Real-time ethylene consumption detection curve for polymerization. (d) Time-dependence studies (yield versus time) of the titanium catalysts (0.5 μmol) at 100 °C, 8 atm.

p.14: Morphology control paragraph: specify Figure 2, efg and Figure 2, abc

Answer: Thanks a lot for your comments. We have specified the description in the revised manuscript.

Additional comments:

In Tables 1, 2, 3 (footnote): catalyst quantity (in μmol) should be replaced by metal quantity (in μmol)

In Tables 2, 3, 5: Incorp. instead of Incrop.

Table 3 title: ethylene polymerization and copolymerization...

Table 4 title: ethylene polymerization and not copolymerization...

P. 12, top: "...the above-mentioned studies on complexes synthesis..."

P. 12, mid: "The generality...makes it easy to support **a mix of** two or more precatalysts..."

In references 4, 5, 8, 10, 12, 16, 19, 41, 50 the names of all authors must be completed; reference 46 is not properly written

Answer: Thanks a lot for your helpful comments. According to the reviewer's comments, we have fixed these issues in the revised manuscript.

In references 4, 5, 8, 10, 12, 16, 19, 41, 50, these references do not include all the authors' names, due to the format requirements of the journal.

Experimental:

P. S3, 2.2 Catalysts preparation: line 2, Catalysts Ni; p. S4, line 9, Catalysts Ti; p. S4, line 19, Catalysts Fe

Page S5, line 5: "...nitrogen, **d**issolve..."

P. S6, 2.3 preparation of supported catalysts, line 7: 1 umol should be 1 μ mol

Answer: Thanks a lot for your helpful comments. According to the reviewer's comments, we have fixed these issues in the revised manuscript.

REVIEWER COMMENTS

Reviewer #1 (Remarks to the Author):

In this revision the authors very carefully addressed the reviewer comments. I feel that they have adequately addressed all of my concerns noted in the previous version of the manuscript, and recommend publication of the manuscript. There are a few minor comments the authors may consider before acceptance, but these are quite minor and I do not see a need to review the manuscript again.

The discussion about Ti-ONa-support and its inability to incorporate polar comonomer is not surprising at all. I believe the authors did this in response to one of the reviewers, but this should probably be removed from the final publication. I have serious doubts that the esters in the polar comonomer are compatible with 2000 equiv of Et₂AlCl (Table 3, entries 15-17)

Do the authors observe reactor fouling at 150-170C polymerization reactions? The polymer can melt off the support under these conditions.

Reviewer #2 (Remarks to the Author):

Strengths: The revised manuscript now includes substitution of data for an Fe catalyst for a phosphino-aryloxide Ni catalyst, the latter of which is active for copolymerizations of ethylene with polar monomers that is a central focus of the study. Time course data on catalyst productivity at high temperatures have also been introduced, and additional spectroscopic data relevant to the nature of the supported catalyst have been added. These data have improved the quality of the study and support the general conclusions.

Weaknesses: My previous criticism about the conceptual novelty of this work, while at odds with the other reviewers, remains. It is still the case that anchoring of catalysts to supports through protic functional groups has years of precedent and success. That the present study demonstrates this strategy in the context of copolymerizations of ethylene with polar comonomers is noteworthy within the polyolefin community but arguably not for a general chemistry readership. The approach is not general (certain organometallic polymerization catalysts most certainly will be unstable toward the presence protic functional groups), modular (a unique ligand synthesis if required for each new catalyst derivative vs homogeneous analogue), or original (citations provided previously).

On balance, this study still seems best suited for a specialized polymer or organometallics journal.

Additional comments:

1) The rebuttal to Reviewer 2, comment 2 is difficult to judge. The graphics here and in Fig S7 and Tables S1 and S2 in the ESI lack structures of the catalysts, which are numerous. It is not practical to search through dozens of citations to find these structures. Irrespective of this complicating issue, plots of activity for copolymerization (e.g., Fig S7, c) provide a clear enough understanding for high activity during copolymerizations with polar monomers: the incorporation of the polar monomer is not exceptional versus other catalysts surveyed. Given the resting state of copolymerization is formed after enchainment of polar alkenes, and chain transfer occurs more readily from this resting state versus simple alkyl-metal intermediates, decreased rate of enchainment should correlate to relatively higher catalyst activity, all other things equal (i.e., catalyst deactivation rate). Polymer molecular weights also tend to

correlate inversely with polar monomer enchainment. In other words, the use of a catalyst that is less able to enchain the polar alkene should be expected to yield better activity and higher polymer MW during the formation of ethylene-rich polymer structures. These new data are appreciated but not particularly persuasive with regards to the overall significance of the study.

2) Data in response to Reviewer 2, comment 4 (parts a-b): these are a nice addition to the revised version that more clearly contrast productivity over time for supported and non-supported catalysts. It is an expected trend considering common biomolecular catalyst decomposition pathways, but these data now more strongly support conclusions about thermal stability.

3) ¹H NMR characterization alone cannot unambiguously establish incorporation of methyl undecenoate versus the presence of a trace high-boiling impurity. The remote ester group resonances are unchanged in free versus incorporated moieties. ¹³C NMR would provide direct characterization of the main chain methine resonance that is characteristic of co-monomer incorporation. Furthermore, the label "b" in Fig S56 is wrongly attributed to a main chain methine rather than the methylene adjacent to an ester group.

4) Labels in a number of NMR spectra obscure key resonances (e.g., "a", "b" in Figs S50-56).

Reviewer #3 (Remarks to the Author):

The manuscript seems to me adequately modified.
I recommend publication in this present form.

Point-by-point response to the reviewers' comments

Reviewer #1 (Remarks to the Author):

In this revision the authors very carefully addressed the reviewer comments. I feel that they have adequately addressed all of my concerns noted in the previous version of the manuscript, and recommend publication of the manuscript. There are a few minor comments the authors may consider before acceptance, but these are quite minor and I do not see a need to review the manuscript again.

Answer: Thank you very much for your comments and support.

The discussion about Ti-ONa-support and its inability to incorporate polar comonomer is not surprising at all. I believe the authors did this in response to one of the reviewers, but this should probably be removed from the final publication. I have serious doubts that the esters in the polar comonomer are compatible with 2000 equiv of Et₂AlCl (Table 3, entries 15-17)

Answer: Thank you very much for your comments and suggestions.

According to the reviewer's comments, we have studied the reaction of methyl 10-undecylenate with 2000 equiv of Et₂AlCl (2M, Hexane) by ¹H NMR. ¹H NMR results indicated interaction between Et₂AlCl and the ester group. However, after adding hydrochloric acid (which is also our treatment method after polymerization), it returns to the initial NMR.

Figure 1. ¹H NMR of the reaction of methyl 10-undecylenate and 2000 equiv of Et₂AlCl (2M, Hexane). (C₆D₆)

This suggest that the interaction/reaction between methyl 10-undecylenate and Et₂AlCl will not affect the copolymerization results and conclusions. To avoid confusion, we have deleted

the studies on Ti-ONa catalyzed copolymerization of ethylene with 10-methyl undecanoate.

Do the authors observe reactor fouling at 150-170°C polymerization reactions? The polymer can melt off the support under these conditions.

Answer: Thanks a lot for your helpful question. During the 150-170 °C polymerization reactions, a small amount of fouling was observed in the reactor.

Reviewer #2 (Remarks to the Author):

Strengths: The revised manuscript now includes substitution of data for an Fe catalyst for a phosphino-aryloxide Ni catalyst, the latter of which is active for copolymerizations of ethylene with polar monomers that is a central focus of the study. Time course data on catalyst productivity at high temperatures have also been introduced, and additional spectroscopic data relevant to the nature of the supported catalyst have been added. These data have improved the quality of the study and support the general conclusions.

Weaknesses: My previous criticism about the conceptual novelty of this work, while at odds with the other reviewers, remains. It is still the case that anchoring of catalysts to supports through protic functional groups has years of precedent and success. That the present study demonstrates this strategy in the context of copolymerizations of ethylene with polar comonomers is noteworthy within the polyolefin community but arguably not for a general chemistry readership. The approach is not general (certain organometallic polymerization catalysts most certainly will be unstable toward the presence protic functional groups), modular (a unique ligand synthesis if required for each new catalyst derivative vs homogeneous analogue), or original (citations provided previously).

On balance, this study still seems best suited for a specialized polymer or organometallics journal.

Answer: Thank you very much for your comments. Our work is different from previously reported works on protic functional groups.

First of all, this work only utilizes protic functional groups as intermediates, which lead to ionic tags. As Reviewer #1 stated: When reading this I was reminded of the strong electrostatic adsorption methods that are very popular in the heterogeneous catalysis community, but to my knowledge have not been reported in the academic literature for olefin polymerization catalysts.

Secondly, this ionic tag strategy enables strong catalyst-support interaction that makes copolymerization possible. For example we recently used the protic functional groups strategy, and reported the generation of heterogeneous nickel catalyst through alcohol induced hydrogen bonding (*Nat. Commun.* **2020**, *11*, 372). The homogeneous nickel catalyst is highly active in ethylene copolymerization with a series of polar comonomers, while the heterogeneous counterpart showed no activity at all in copolymerization. These results clearly demonstrated the advantages of this heterogenization strategy in generating supported late transition metal catalysts, especially for the purpose of copolymerization reactions.

Third, the protic functional groups were only as intermediates that leading to ionic tags, making this approach quite general for many different kinds of olefin polymerization catalysts. In this

work, we have deliberately demonstrated suitability for phosphine based (phosphinophenolate nickel) and imine based (phenoxy-imine titanium, and pyridine-diimine iron) olefin polymerization catalysts. These two types of ligands (phosphine and imine) actually represent the majority types of ligands for olefin polymerization catalysts (see the chemical structures of catalysts shown in Table S1, S2, S3 and S4).

Additional comments:

1) The rebuttal to Reviewer 2, comment 2 is difficult to judge. The graphics here and in Fig S7 and Tables S1 and S2 in the ESI lack structures of the catalysts, which are numerous. It is not practical to search through dozens of citations to find these structures. Irrespective of this complicating issue, plots of activity for copolymerization (e.g., Fig S7, c) provide a clear enough understanding for high activity during copolymerizations with polar monomers: the incorporation of the polar monomer is not exceptional versus other catalysts surveyed. Given the resting state of copolymerization is formed after enchaining polar alkenes, and chain transfer occurs more readily from this resting state versus simple alkyl-metal intermediates, decreased rate of enchainment should correlate to relatively higher catalyst activity, all other things equal (i.e., catalyst deactivation rate). Polymer molecular weights also tend to correlate inversely with polar monomer enchainment. In other words, the use of a catalyst that is less able to enchain the polar alkene should be expected to yield better activity and higher polymer MW during the formation of ethylene-rich polymer structures. These new data are appreciated but not particularly persuasive with regards to the overall significance of the study.

Answer: Thank you very much for your comments and suggestions.

First, we have provided the chemical structures for all the catalysts cited in Table S1, S2, S3 and S4.

Second, please note that Figure S7c provides comparison of polymer molecular weight and comonomer incorporation between representative reported **homogeneous nickel/palladium catalysts** and **heterogeneous Ni-ONa-MgO/Ni-F-ONa-MgO** in the copolymerization of ethylene with methyl acrylate or tert-butyl acrylate. Please note that our heterogeneous catalysts can achieve higher comonomer incorporation. We have provided two additional data points (from Table 4, entry 5 and entry 6) to demonstrate that under similar comonomer incorporation level, our catalyst showed higher activity and higher molecular weight. With ca. 7 mol% of tert-butyl acrylate incorporation (from Table 4, entry 5 and entry 6; 6.6 and 7.4 mol% incorporation, respectively), these heterogeneous catalysts still showed higher activity than reported **homogeneous catalysts**.

Third, the differences are larger for the comparison of our heterogeneous catalysts versus

previously reported **heterogeneous** catalysts (Figure S7d).

Fourth, the reviewer is correct that it is a formidable challenge to improve one of the three properties (molecular weight, incorporation, and activity) without sacrificing the other two. However, our studies showed that the supporting of the nickel catalysts can simultaneously improve all three parameters (**Ni-ONa-MgO** versus **Ni-ONa**; Table 2, entry 7 versus entry 2). This further demonstrate the novelty and superior properties of this ionic tag strategy.

2) Data in response to Reviewer 2, comment 4 (parts a-b): these are a nice addition to the revised version that more clearly contrast productivity over time for supported and non-supported catalysts. It is an expected trend considering common biomolecular catalyst decomposition pathways, but these data now more strongly support conclusions about thermal stability.

Answer: Thank you very much for your comments.

3) ¹H NMR characterization alone cannot unambiguously establish incorporation of methyl undecenoate versus the presence of a trace high-boiling impurity. The remote ester group resonances are unchanged in free versus incorporated moieties. ¹³C NMR would provide direct characterization of the main chain methine resonance that is characteristic of co-monomer incorporation. Furthermore, the label “b” in Fig S56 is wrongly attributed to a main chain methine rather than the methylene adjacent to an ester group.

Answer: Thanks a lot for your comments. According to the reviewer’s comments, we provided the ¹³C NMR analysis of methyl 10-undecylenate/ethylene copolymer (Table 4, Entry 9). The ¹³C NMR showed that the methyl 10-undecenoate comonomer was incorporated.

According to the reviewer’s comments, we have checked the labels in the NMR spectra.

Figure 2. ¹³C NMR of the copolymer from methyl 10-undecylenate/ethylene copolymer (Table 4, Entry 9). (C₂D₂Cl₄, 120°C)

4) Labels in a number of NMR spectra obscure key resonances (e.g., “a”, “b” in Figs S50-56).

Answer: Thanks for catching this. We have fixed this issue.

Reviewer #3 (Remarks to the Author):

The manuscript seems to me adequately modified.

I recommend publication in this present form.

Answer: Thank you very much for your comments and support.

REVIEWER COMMENTS

Reviewer #1 (Remarks to the Author):

The authors addressed my very minor comments in this revision, and I recommend publication of the manuscript.

Reviewer #2 (Remarks to the Author):

The authors in the new rebuttal continue to focus on ranking the performance of these catalysts versus others in prior studies. Firstly, these assertions that the differences noted are sufficient in magnitude warrant publication in a general chemistry journal continue to remain unpersuasive to this reviewer. Secondly, it continues to skirt what was and is still my principal concern: This work centers on a supported late metal catalyst that incorporates polar monomers during insertion copolymerization. Citations given in prior review highlight past work (#1) using distal protic functional groups to anchor late metal catalysts onto supports. Refs 37-40 demonstrate (#2) copolymerizations with polar monomers using supported late metal catalysts. The present work applies catalysts of type #1 to applications of type #2. The rebuttals have not ameliorated this concern that the union of the above two concepts does not represent a conceptual advance exceeding a threshold expected for this journal.

Responses to new rebuttal comments:

1. Answer: Thank you very much for your comments. Our work is different from previously reported works on protic functional groups.

First of all, this work only utilizes protic functional groups as intermediates, which lead to ionic tags. As Reviewer #1 stated: When reading this I was reminded of the strong electrostatic adsorption methods that are very popular in the heterogeneous catalysis community, but to my knowledge have not been reported in the academic literature for olefin polymerization catalysts

>> *I disagree, still. Citations provided in the first review demonstrate the concept of anchoring homogeneous catalysts onto silica through distal protic functional groups.*

2. Secondly, this ionic tag strategy enables strong catalyst-support interaction that makes copolymerization possible. For example we recently used the protic functional groups strategy, and reported the generation of heterogeneous nickel catalyst through alcohol induced hydrogen bonding (*Nat. Commun.* **2020**, *11*, 372). The homogeneous nickel catalyst is highly active in ethylene copolymerization with a series of polar comonomers, while the heterogeneous counterpart showed no activity at all in copolymerization. These results clearly demonstrated the advantages of this heterogenization strategy in generating supported late transition metal catalysts, especially for the purpose of copolymerization reactions.

>> *The authors provide citations (Ref 37-40) countering their own assertions that (1) the present work represents a novel demonstration of insertion copolymerization with polar alkenes using supported late metal catalysts and (2), that the “ionic tag strategy” is essential.*

3. Third, the protic functional groups were only as intermediates that leading to ionic tags, making this approach quite general for many different kinds of olefin polymerization catalysts. In this work, we have deliberately demonstrated suitability for phosphine based (phosphinophenolate nickel) and imine based (phenoxy-imine titanium, and pyridine-diimine iron) olefin polymerization catalysts. These two types of ligands (phosphine and imine) actually represent the majority types of ligands for olefin polymerization catalysts (see the chemical structures of catalysts shown in Table S1, S2, S3 and S4).

>> *As noted previously this notion is undermined by the fact that each new catalyst with an “ionic tag” requires a bespoke ligand synthesis versus the respective known homogeneous catalyst. That is neither general nor expedient. The comments above are redundant and do not rebut this concern.*

4. First, we have provided the chemical structures for all the catalysts cited in Table S1, S2, S3 and S4.

Second, please note that Figure S7c provides comparison of polymer molecular weight and comonomer incorporation between representative reported **homogeneous nickel/palladium catalysts** and **heterogeneous** Ni-ONa-MgO/Ni-F-ONa-MgO in the copolymerization of ethylene with methyl acrylate or tert-butyl acrylate. Please note that our heterogeneous catalysts can achieve higher comonomer incorporation. We have provided two additional data points (from Table 4, entry 5 and entry 6) to demonstrate that under similar comonomer incorporation level, our catalyst showed higher activity and higher molecular weight. With ca. 7 mol% of tert-butyl acrylate incorporation (from Table 4, entry 5 and entry 6; 6.6 and 7.4 mol% incorporation, respectively), these heterogeneous catalysts still showed higher activity than reported **homogeneous catalysts**.

>> *The comparison the authors are making here is ambiguous. "Higher than reported homogeneous catalysts" Which catalysts? What citations? The generality of this claim precludes evaluation of its veracity.*

5. Third, the differences are larger for the comparison of our heterogeneous catalysts versus previously reported **heterogeneous** catalysts (Figure S7d).

>> *It is difficult to make substantive conclusions from "apples to oranges" data such as these, which are derived from studies conducted under non standardized conditions. The reasons for variations in catalyst performance are potentially complex and manifold; it would be unfair to judge this study's significance, irrespective of whether it represents a flattering or unfavorable comparison, without clearer understanding of the reasons for discrepancies in performance.*

6. Fourth, the reviewer is correct that it is a formidable challenge to improve one of the three properties (molecular weight, incorporation, and activity) without sacrificing the other two. However, our studies showed that the supporting of the nickel catalysts can simultaneously improve all three parameters (**Ni-ONa-MgO** versus **Ni-ONa**; Table 2, entry 7 versus entry 2). This further demonstrates the novelty and superior properties of this ionic tag strategy.

>> *I disagree this has been demonstrated to any convincing degree based on the data in Table 2 because it is a comparison to an arbitrary control. What is the justification the performance of Ni-ONa represents an informative representation of existing state-of-the-art homogeneous catalysts writ large? On the contrary, contemporary examples of (phosphino-aryloxy)Ni catalysts (see, for example: Agapie JACS 2021, 143, 6516; activity up to 660 kg(mol·h)⁻¹, Mw up to 100 kDa, acrylate incorporation up to 12 mol%) suggest the data in Table 2 are not remarkable compared to existing catalysts of this general class.*

We want to thank the reviewers for the efforts in accessing our manuscript. These comments have helped us a lot to improve our work.

We have provided detailed responses to the reviewer's comments.

The reviewer did not raise any issues on the technical or scientific aspects. The reviewer basically has three issues on our work: novelty, generality, and performance.

1. **Novelty.**

Both reviewer 1 and reviewer 3 supported the novelty of this work. As Reviewer #1 stated: When reading this I was reminded of the strong electrostatic adsorption methods that are very popular in the heterogeneous catalysis community, but to my knowledge have not been reported in the academic literature for olefin polymerization catalysts.

The reviewer stated "The present work applies catalysts of type #1 to applications of type #2." Previous works (#1) demonstrated using distal protic functional groups to anchor late metal catalysts on supports, but did not study polar monomer copolymerization. Previous works (#2) demonstrated polar monomer copolymerization studies using supported late metal catalysts (prepared using other supporting strategies). Our work is not "applies catalysts of type #1 to applications of type #2" for the following reasons (see below for details).

2. **Generality.**

The "ionic tag" DOES NOT require a bespoke ligand synthesis. The synthetic procedures for the OM functionalized ligands are basically the same as the counterparts without OM tags. The only difference lies in the starting materials bearing OH/OM tags (which are mostly commercially available). Most importantly, the OH/OM tags do not interfere with the metal complex syntheses.

As requested by the reviewer, we have provided the chemical structures of previously reported olefin polymerization catalysts (see the chemical structures of catalysts shown in Table S1, S2, S3 and S4). The two types of ligands (phosphine and imine) actually represent the majority types of ligands. In this work, we have deliberately demonstrated suitability for phosphine based (phosphinophenolate nickel) and imine based (phenoxy-imine titanium, and pyridine-diimine iron) olefin polymerization catalysts.

3. **Performance.**

As requested by the reviewer, we have put the most active catalysts previously reported together (Figure S7, Table S1, S2, S3 and S4). The comparison clearly demonstrated the high performance of our new catalysts.

We want to thank the reviewer for providing a specific example, which we have actually cited and included in our comparison (reference 51 in supporting information). Comparing with the performance in ethylene/*tert*-butyl acrylate copolymerization (activity up to 660 kg(mol·h)⁻¹, Mw up to 100 kDa, acrylate incorporation up to 12 mol%), the corresponding performance of our catalyst is much better: activity up to 4100 kg(mol·h)⁻¹, Mw up to 1400 kDa, acrylate incorporation up to 7.4 mol% (currently, we are only using 0.2M comonomer concentration, this value can be easily increased by using higher comonomer

concentration, low ethylene pressure, or sterically less bulky ligand).

Point-by-point response to the reviewers' comments

Citations given in prior review highlight past work (#1) using distal protic functional groups to anchor late metal catalysts onto supports. Refs 37-40 demonstrate (#2) copolymerizations with polar monomers using supported late metal catalysts. The present work applies catalysts of type #1 to applications of type #2. The rebuttals have not ameliorated this concern that the union of the above two concepts does not represent a conceptual advance exceeding a threshold expected for this journal.

Answer: see responses to 1.

1. Answer: Thank you very much for your comments. Our work is different from previously reported works on protic functional groups. First of all, this work only utilizes protic functional groups as intermediates, which lead to ionic tags. As Reviewer #1 stated: When reading this I was reminded of the strong electrostatic adsorption methods that are very popular in the heterogeneous catalysis community, but to my knowledge have not been reported in the academic literature for olefin polymerization catalysts

>> I disagree, still. Citations provided in the first review demonstrate the concept of anchoring homogeneous catalysts onto silica through distal protic functional groups.

Answer: Previous works (#1) demonstrated using distal protic functional groups to anchor late metal catalysts on supports, but did not study polar monomer copolymerization. Previous works (#2) demonstrated polar monomer copolymerization studies using supported late metal catalysts (prepared using other supporting strategies). Our work is not “applies catalysts of type #1 to applications of type #2” for the following reasons:

- i. First of all, “applies catalysts of type #1 to applications of type #2” actually DOES NOT work for polar monomer copolymerization. Our previous work (Nat. Commun. 2020, 11, 372) utilizes distal protic functional group to anchor nickel catalyst (fall into type #1 works), and the resulting heterogeneous catalyst showed no activity in polar monomer copolymerization.
- ii. The utilization of distal protic functional groups (hydroxyl group: OH) is completely different from the present work, which utilizes OM (M=Na, K, etc). For OH to anchor on solid support, the support needs to be pretreated with aluminum cocatalyst (shown in previous type #1 references). For OM to anchor on solid support, no pretreatment of the solid support is required.
- iii. The OM-support interaction is much stronger than OH-support interaction, which is crucial for polar monomer copolymerization.

2. Secondly, this ionic tag strategy enables strong catalyst-support interaction that makes copolymerization possible. For example we recently used the protic functional groups strategy, and reported the generation of heterogeneous nickel catalyst through alcohol induced hydrogen bonding (Nat. Commun. 2020, 11, 372). The homogeneous nickel catalyst is highly active in ethylene copolymerization with a series of polar comonomers, while the heterogeneous counterpart showed no activity at all in copolymerization. These results clearly demonstrated the advantages of this heterogenization strategy in generating supported late transition metal

catalysts, especially for the purpose of copolymerization reactions.

>> The authors provide citations (Ref 37-40) countering their own assertions that (1) the present work represents a novel demonstration of insertion copolymerization with polar alkenes using supported late metal catalysts and (2), that the “ionic tag strategy” is essential.

Answer: The novelty of the work is supported by Reviewer 1 and Reviewer 3. (1) The present work represents a novel concept because previous type #1 works utilizes OH group to anchor catalyst, and previous type #2 works utilizes other supporting strategies for polar monomer copolymerization. Simply applying type #1 works to type #2 works does not work for polar monomer copolymerization. Our work is different from either type #1 works or type #2 works. (2) The “ionic tag strategy” is essential because the utilization of OH tag strategy (type #1 works) does not work for polar monomer copolymerization (Nat. Commun. 2020, 11, 372).

3. Third, the protic functional groups were only as intermediates that leading to ionic tags, making this approach quite general for many different kinds of olefin polymerization catalysts. In this work, we have deliberately demonstrated suitability for phosphine based (phosphinophenolate nickel) and imine based (phenoxy-imine titanium, and pyridine-diimine iron) olefin polymerization catalysts. These two types of ligands (phosphine and imine) actually represent the majority types of ligands for olefin polymerization catalysts (see the chemical structures of catalysts shown in Table S1, S2, S3 and S4).

>> As noted previously this notion is undermined by the fact that each new catalyst with an “ionic tag” requires a bespoke ligand synthesis versus the respective known homogeneous catalyst. That is neither general nor expedient. The comments above are redundant and do not rebut this concern.

Answer: The “ionic tag” DOES NOT require a bespoke ligand synthesis. The synthetic procedures for the OM functionalized ligands are basically the same as the counterparts without OM tags. The only differences lie in the starting materials bearing OH/OM tags (which are mostly commercially available). Most importantly, the OH/OM tags do not interfere with the metal complex syntheses.

As requested by the reviewer, we have provided the chemical structures of previously reported olefin polymerization catalysts (see the chemical structures of catalysts shown in Table S1, S2, S3 and S4). The two types of ligands (phosphine and imine) actually represent the majority types of ligands. In this work, we have deliberately demonstrated suitability for phosphine based (phosphinophenolate nickel) and imine based (phenoxy-imine titanium, and pyridine-diimine iron) olefin polymerization catalysts.

4. First, we have provided the chemical structures for all the catalysts cited in Table S1, S2, S3 and S4. Second, please note that Figure S7c provides comparison of polymer molecular weight and comonomer incorporation between representative reported homogeneous nickel/palladium catalysts and heterogeneous Ni-ONa-MgO/Ni-F-ONa-MgO in the copolymerization of ethylene with methyl acrylate or tert-butyl acrylate. Please note that our heterogeneous catalysts can achieve higher comonomer incorporation. We have provided two additional data points (from Table 4, entry 5 and entry 6) to demonstrate that under similar comonomer incorporation level, our catalyst showed higher activity and higher molecular weight. With ca. 7 mol% of tert-butyl acrylate incorporation (from Table 4, entry 5 and entry 6; 6.6 and 7.4 mol%

incorporation, respectively), these heterogeneous catalysts still showed higher activity than reported homogeneous catalysts.

>> The comparison the authors are making here is ambiguous. “Higher than reported homogeneous catalysts” Which catalysts? What citations? The generality of this claim precludes evaluation of its veracity.

Answer: The catalysts are all the catalysts shown in Figure S7, Table S1, S2, S3 and S4, which are the most active catalysts reported in literature.

5. Third, the differences are larger for the comparison of our heterogeneous catalysts versus previously reported heterogeneous catalysts (Figure S7d).

>> It is difficult to make substantive conclusions from “apples to oranges” data such as these, which are derived from studies conducted under non standardized conditions. The reasons for variations in catalyst performance are potentially complex and manifold; it would be unfair to judge this study’s significance, irrespective of whether it represents a flattering or unfavorable comparison, without clearer understanding of the reasons for discrepancies in performance.

Answer: The differences comparison between our catalysts versus previously reported catalysts is actually required by reviewer 2.

6. Fourth, the reviewer is correct that it is a formidable challenge to improve one of the three properties (molecular weight, incorporation, and activity) without sacrificing the other two. However, our studies showed that the supporting of the nickel catalysts can simultaneously improve all three parameters (Ni-ONa-MgO versus Ni-ONa; Table 2, entry 7 versus entry 2). This further demonstrate the novelty and superior properties of this ionic tag strategy.

>> I disagree this has been demonstrated to any convincing degree based on the data in Table 2 because it is a comparison to an arbitrary control. What is the justification the performance of Ni-ONa represents an informative representation of existing state-of-the-art homogeneous catalysts writ large? On the contrary, contemporary examples of (phosphino-aryloxy)Ni catalysts (see, for example: Agapie JACS 2021, 143, 6516; activity up to 660 kg(mol·h)⁻¹, Mw up to 100 kDa, acrylate incorporation up to 12 mol%) suggest the data in Table 2 are not remarkable compared to existing catalysts of this general class.

Answer: Thanks lot for providing a specific example, which we have actually cited and included in our comparison (reference 51 in supporting information).

Comparing with the performance in ethylene/*tert*-butyl acrylate copolymerization (activity up to 660 kg(mol·h)⁻¹, Mw up to 100 kDa, acrylate incorporation up to 12 mol%), the corresponding performance of our catalyst is much better: activity up to 4100 kg(mol·h)⁻¹, Mw up to 1400 kDa, acrylate incorporation up to 7.4 mol% (currently, we are only using 0.2M comonomer concentration, this value can be easily increased by using higher comonomer concentration, low ethylene pressure, or sterically less bulky ligand).